# Representation of basal melting at the grounding line in ice flow models

Hélène Seroussi[1] and Mathieu Morlighem[2]

[1]Jet Propulsion Laboratory - California Institute of Technology, Pasadena, CA 91109, USA
[2]Department of Earth System Science, University of California Irvine, Irvine, CA 92697, USA

*Correspondence to:* Helene Seroussi (Helene.Seroussi@jpl.nasa.gov)

**Abstract.** While a lot of attention has been given to the numerical implementation of grounding lines and basal friction in the grounding zone, little has been done about the impact of the numerical treatment of ocean-induced basal melting in this region. Several strategies are currently being employed in the ice sheet modeling community, and the resulting grounding line dynamics may differ strongly, which ultimately adds significant uncertainty to the projected contribution of marine ice sheets to sea level rise. We investigate here several implementations of basal melt parameterization on partially floating elements in a finite element framework, based on the Marine Ice Sheet-Ocean Model Intercomparison Project (MISOMIP) setup: (1) melt applied only to entirely floating elements, (2) melt applied over the entire elements that are crossed by the grounding line, and (3) melt integrated partially over the floating portion of a finite element using two different sub-element integration methods. All methods converge towards the same state when the mesh resolution is fine enough. However, (2) and (3) will systematically overestimate the rate of grounding line retreat in coarser resolutions, while (1) converges faster to the solution in most cases. The differences between sub-element parameterizations are exacerbated for experiments with large melting rates in the vicinity of the grounding line and for a Weertman sliding law. As most real-world simulations use horizontal mesh resolutions of several hundreds of meters at best, and large melt rates are generally present close to the grounding lines, we recommend not using (3) to avoid overestimating the rate of grounding line retreat and to carefully assess the impact of mesh resolution and sub-element melt parameterizations on all simulation results.

## 1   Introduction

Basal melt under floating ice tongues is important as it is one of the main factors driving the current increase in ice discharge in West Antarctica (e.g. Pritchard et al., 2012). Changes in basal melt impact ice shelf thickness, and thinning leads to a reduction of ice shelf buttressing, thereby leading to an acceleration of the ice streams feeding it. This acceleration is responsible for the dynamic thinning of the ice upstream of the grounding line, eventually leading to grounding line retreat, which causes to a further increase in ice speed, and therefore ice discharge. Accurate representation of ice shelf ocean-induced melt in ice flow models is therefore critical. This remains an active field of research as observations of basal melt remain scarce, and new parameterizations are starting to emerge (Lazeroms et al., 2018; Reese et al., 2017).

Over the past decade, the ice sheet modeling community has made tremendous progress in terms of representation of grounding line dynamics in ice sheet models. Model intercomparisons have shown that lateral stress and high mesh resolution (below 2 km) in the grounding zone are required to accurately capture the behavior of the grounding line (Pattyn et al., 2012, 2013). New sub-element parameterizations of grounding line position and the representation of basal friction in partially floating elements showed promising results for both flow band and plan view models (Pattyn et al., 2006; Gladstone et al., 2010; Seroussi et al., 2014a; Feldmann et al., 2014), as they relaxed the mesh resolution requirements in this region. These studies, however, are all based on ideal geometries and completely ignored basal melt under floating ice (i.e., no melt is applied under floating ice). In reality, melt can be strong, especially in the vicinity of the grounding line, where it can reach $\sim$100 m/yr (Dutrieux et al., 2013; Rignot et al., 2013; Berger et al., 2017). Several studies have showed that for the same melt parameterization, the choice of numerical implementation of melt has a strong impact on model results for both projections of the West Antarctic Ice Sheet (Cornford et al., 2016; Arthern and Williams, 2017) and idealized glaciers (Gladstone et al., 2017). This problem has however not been fully investigated or quantified yet, and it remains unclear what parameterizations should be employed in partially floating elements.

We investigate these questions here by using different numerical implementations of basal melting in partially floating elements and two friction laws on a setup similar to the Marine Ice Sheet-Ocean Model Intercomparison Project (MISOMIP) (Asay-Davis et al., 2016). We first summarize the model setup and detail the four different parameterizations of basal melt in elements partially floating and partially grounded. We then describe the experiments used to test these parameterizations. We present the results, discuss their impact on the modeling of grounding line evolution and conclude on the relevance of using sub-element parameterizations of ocean-induced melt under ice shelves.

## 2  Model

We use the Ice Sheet System Model (ISSM, Larour et al., 2012) to simulate the ice flow of an idealized case representative of outlet glaciers in West Antarctica (Asay-Davis et al., 2016). The model setup is identical to the one described in Asay-Davis et al. (2016) that we briefly summarize here. All the parameters are identical to their description, except where specified otherwise.

The experiments simulate a glacier in a marine terminating confined valley, with a bedrock lying between -720 and 350 m as shown in Fig. 1a. The accumulation is uniform over the domain and set to 0.3 m/yr. Basal melting is applied under floating ice, with a different magnitude depending on the experiments. The model domain extends between 0 and 640 km, and between 0 and 80 km in the $x$ and $y$ direction, respectively. This domain is discretized using a triangular mesh with resolutions of 2 km, 1 km, 500 m, 250 m, and 125 m resulting in meshes with a number of elements varying from 28,000 to 1,745,000. All mesh resolutions are spatially uniform, except in the case of the 125 m resolution mesh for which the model resolution is 125 m only in the portion of the domain located between $x = 300$ km and $x = 600$ km (i.e. where we expect to see the grounding line), the resolution is otherwise 1 km for $x < 200$ km, and 500 m for the rest of the domain.

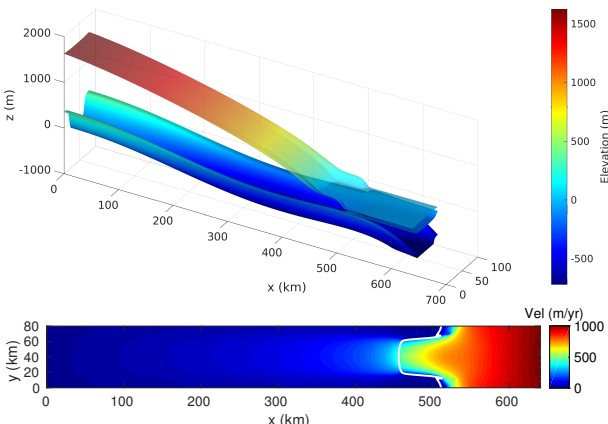

**Figure 1.** Model domain and initial steady-state geometry for the 125 m resolution mesh with a Weertman sliding law. (a) Bedrock elevation and initial steady-state ice surface and basal elevation (Note the vertical exaggeration). (b) Initial steady-state velocity (in m/yr). The white line shows the initial grounding line position.

The two-dimensional Shelfy-Stream Approximation (MacAyeal, 1989) is used as an approximation of the full-Stokes equations to solve the stress balance equations and the grounding line position is determined assuming hydrostatic equilibrium. The ice rheology is spatially uniform in the domain and follows Glen's flow law with a rate factor, $A$, equal to $2.0 \times 10^{-17}$ Pa$^{-3}$ yr$^{-1}$, equivalent to an ice temperature of about -9°C. Boundary conditions are a no slip condition at $x = 0$ km, a free-slip condition at $y = 0$ and $y = 80$ km, and a fixed ice front at $x = 640$ km. We test here two different friction laws. The first one is a power sliding law, following Weertman (1957):

$$\boldsymbol{\tau}_b = -\beta^2 \|\mathbf{u}_b\|^{1/m-1} \mathbf{u}_b \tag{1}$$

with $\boldsymbol{\tau}_b$ the basal stress, $\mathbf{u}_b$ the basal velocity vector, $m = 3$ and $\beta^2$ the friction coefficient uniform in space and equal to $1.0 \times 10^4$ Pa m$^{-1/3}$ yr$^{1/3}$. This friction law induces a sharp discontinuity in basal friction at the grounding line that is not realistic and not appropriate for problems investigating grounding line evolution, but remains nevertheless widely used in the community (Brondex et al., 2017).

The second sliding law is a modified power law designed to prevent the basal traction to exceed a fraction of the effective pressure, proposed by Tsai et al. (2015):

$$\boldsymbol{\tau}_b = -\min \left( \alpha^2 N, \beta^2 \|\mathbf{u}_b\|^{1/m} \right) \|\mathbf{u}_b\|^{-1} \mathbf{u}_b \tag{2}$$

with $\alpha^2 = 0.5$ and $N$ the effective pressure at the ice base, assuming a perfect connectivity of the subglacial hydrologic system with the ocean.

The representation of basal friction at the grounding line is the same in all experiments, and follows the SEP2 parameterization of Seroussi et al. (2014a). It has been shown that this parameterization is satisfactory to capture grounding line dynamics, as it converges faster to the solution as the mesh resolution increases compared to other methods.

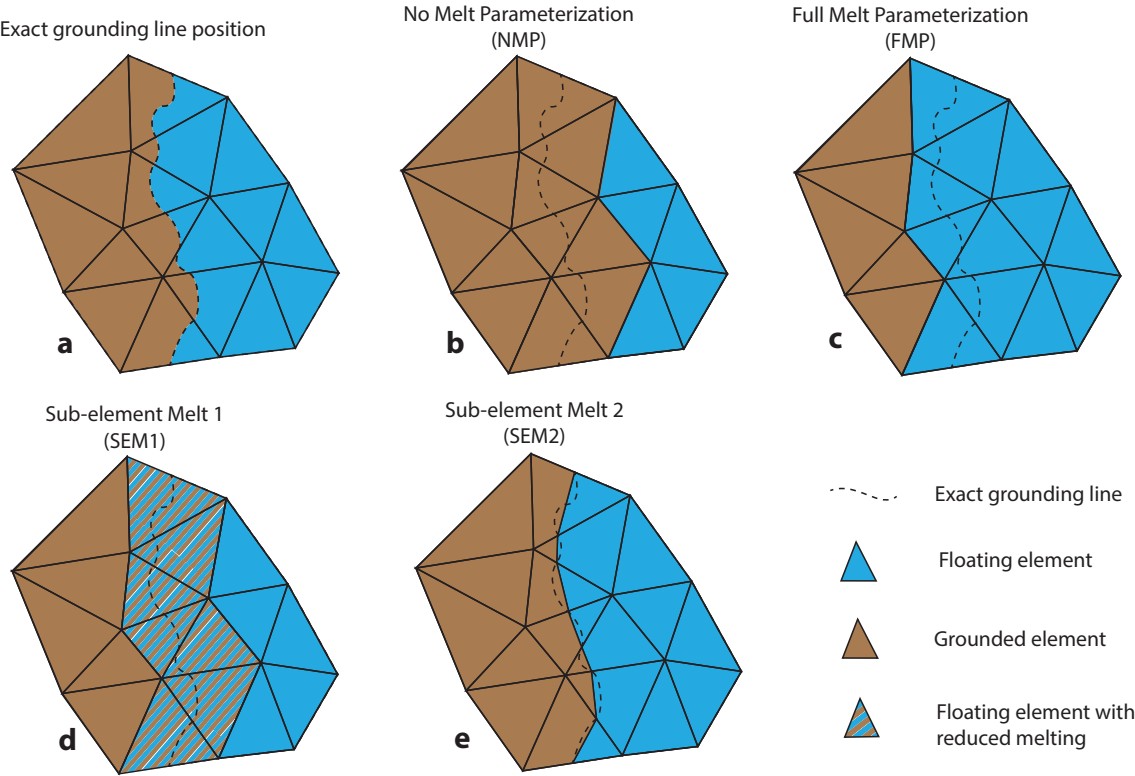

**Figure 2.** Grounding line discretization. Grounding line exact location (a), No Melt Parameterization (NMP, b), Full Melt Parameterization (FMP, c), Sub-Element Melt 1 (SEM1, d), and Sub-Element Melt 2 (SEM2, e)

In this study, we use the same methodology as Seroussi et al. (2014a), but apply it to sub-element melting parameterizations in elements partially floating and partially grounded. Figure 2 shows the four different parameterizations adopted in this study. In the case of the 'Full Melt parameterization' (FMP), melt is applied everywhere over all partially floating elements, regardless of the exact position of the grounded line, while in the 'No Melt Parameterization' (NMP), there is no melt applied to any area of the partially floating elements. The last two cases use a sub-element parameterization. In the 'Sub-Element Melt 1' (SEM1), melt is applied to the entire area of partly floating element, but the magnitude of the melt is reduced by the fraction area of the floating ice in the element, so that the total melt applied is proportional to the floating ice area. In the 'Sub-Element Melt 2' (SEM2) parameterization, the ocean induced melt rate is integrated exactly over the floating part of the element in the mass transport equation, so that melt rate is only applied to the floating part of the element.

Testing two sliding laws, four melt parameterizations and five mesh resolutions results in a total of 40 different configurations. The same experiments are performed on each of these configurations.

## 3 Experiments

We first run every configuration to a steady-state ice stream without any melt. The initial ice thickness is equal to 1 m and the ice stream grows over several tens of thousands of years (at least 50,000 years) in response to surface mass balance accumulation, while no basal melting is applied under floating ice. This steady-state is therefore independent of the sub-element basal melt parameterization applied. Convergence of the solution to the steady-state is discussed in the analysis of Experiment 0 in Section 4.

Starting from this steady-state, three transient experiments with varying ice shelf basal melting conditions are performed for a period of 100 years. In Experiment 0, no basal melting is applied under floating ice, similar to the steady-state initialization of the model. Experiment 0 is therefore mainly designed to check the initial steady-state. Basal melting is applied under floating ice in Experiment 1 and Experiment 2, and we assess the impact of the melt parameterization, model resolution and sliding laws on the glacier evolution. Experiment 1 is similar to the MISMIP+ *Ice1r* experiment in Asay-Davis et al. (2016): basal melting varies spatially and represents a balance between the latent heat of melting and a parameterized ocean turbulent heat flux:

$$m_i = \Omega \tanh\left(\frac{H_c}{H_{c0}}\right) \max\left(z_0 - z_d, 0\right) \tag{3}$$

with $\Omega$ a coefficient equal to 0.2 $\text{yr}^{-1}$, $H_c$ the water column thickness, $z_d$ the ice shelf basal elevation, $z_0$ the depth above which the melt rate is equal to zero (100 m), and $H_{c0}$ a constant equal to 75 m (see also equations (12)-(17) in Asay-Davis et al. (2016) for the derivation of this parameterization).

Experiment 2 is based on a basal melt under floating ice that varies linearly with depth, with a maximum melt magnitude of 30 m/yr in the deepest part, where the ice base is at or below 500 m below sea level, and that linearly decreases to 0 m/yr melt for ice base equal to 50 m below sea level. There is therefore no melt when the ice base is above 50 m below sea level:

$$m_i = \begin{cases} 0 \text{ m/yr,} & \text{if } z_d > -50 \text{ m} \\ -1/15\left(z_d + 50\right) \text{ m/yr,} & \text{if } -500 < z_d < -50 \text{ m} \\ 30 \text{ m/yr,} & \text{if } z_d < -500 \text{ m} \end{cases} \tag{4}$$

with $z_d$ the ice shelf basal elevation. This experiment simulates ice shelves resting in warm waters, similarly to what has been observed in the Amundsen or Bellingshausen sea areas (e.g. Dutrieux et al., 2013; Rignot et al., 2013) and used in previous modeling experiments (e.g. Favier et al., 2014; Joughin et al., 2014; Seroussi et al., 2014b, 2017).

Experiments 0, 1 and 2 are all run for 100 years. We use the following convention to refer to the different experiments. For the steady-state (SS) and Experiment 0, names are as follows: EXP_sliding_resolution, where EXP is the number of the experiment (SS or EXP0), the sliding refers to the sliding law (Weertman or Tsai), and 'resolution' is the mesh resolution (2 km, 1 km, 500 m, 250 m, or 125 m), e.g., EXP0_Weertman_250m. For Experiment 1 and Experiment 2, the names are similar: EXP_sliding_resolution_SEM, except that we add SEM, the sub-element melt parameterization at the grounding line

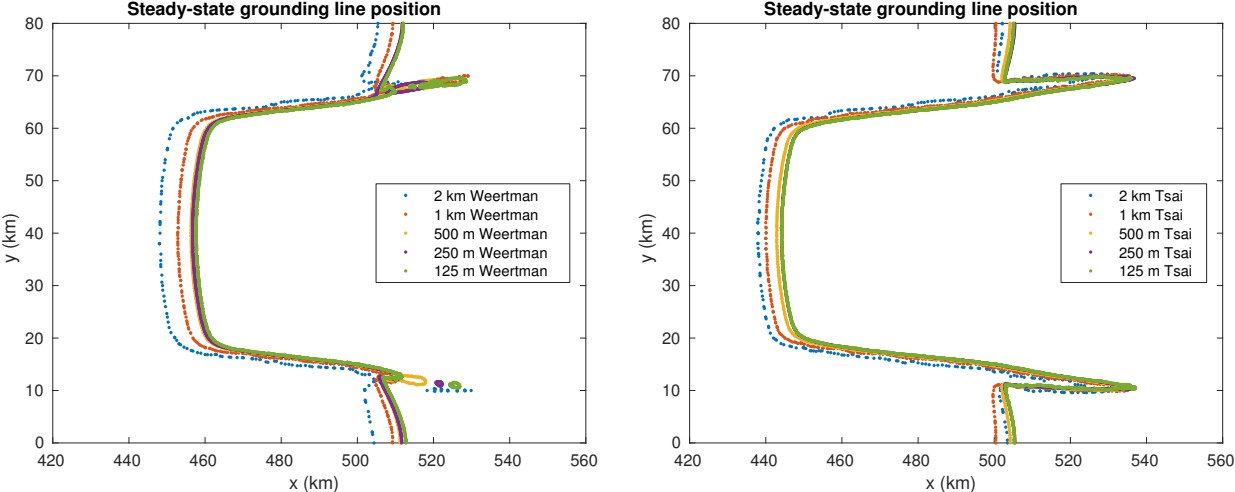

**Figure 3.** Steady-state grounding line positions for the Weertman (left) and Tsai (right) friction law for the 2 km (blue), 1 km (red), 500 m (yellow), 250 m (purple) and 125 m (green) mesh resolutions. 250 m and 125 m resolution grounding are superimposed for the Tsai friction law.

(NMP, FMP, SEM1 or SEM2), e.g., EXP1_Weertman_250m_SEM1, as the results of these simulations now depend on the sub-element melt parameterization adopted.

## 4 Results

Figure 1 shows the initial steady-state configuration for SS_Weertman_125m. Its geometry is shown in Fig. 1a, and the velocity and grounding line in Fig. 1b. The grounding line position varies between 458 km in the centerline of the glacier and 528 km on its sides; the ice velocity is maximum at the ice front, reaching 1012 m/yr. This configuration is comparable to previous results based on the same geometry (Gudmundsson et al., 2012; Gudmundsson, 2013; Asay-Davis et al., 2016). The mesh resolution and the type of basal sliding law both impact the grounding line position as shown in Fig. 3. The grounding line position on the glacier centerline varies between 438 km for SS_Tsai_2km and 458 km for SS_Weertman_125m, with a larger spread between the different resolutions for the Tsai friction law (9.6 km) than for the Weertman friction law (6.2 km) (Fig. 3 and Table 1).

Experiment 0 is mostly designed to ensure that the model has reached a steady-state, as no melt is applied, similar to the initial steady-state. The ice mass above floatation (Fig. 4a) remains constant over the 100-year simulation for the 10 configurations, while the grounded ice area (Fig. 4b) experiences small oscillations with small oscillations, especially for the Weertman sliding law. Such oscillations, that average to zero change in the grounded area over time, have been noted by Asay-Davis et al. (2016) and are orders of magnitude smaller than the changes simulated in Experiment 1 and Experiment 2. Figure 4 confirms that sub-kilometer resolution is needed to accurately capture the grounding line positions, similarly to what has been suggested by previous studies (e.g., Vieli and Payne (2005); Gladstone et al. (2010); Pattyn et al. (2012, 2013); Feldmann et al. (2014);

**Table 1.** Steady-state grounding line position in the glacier centerline and volume above floatation (VAF)

| Friction law | Resolution | GL$(y = 40$ km$)$ | VAF (Gt) |
|---|---|---|---|
| Weertman | 2 km | 448.0 km | 46327 |
| Weertman | 1 km | 452.8 km | 47044 |
| Weertman | 500 m | 456.3 km | 47540 |
| Weertman | 250 m | 456.6 km | 47674 |
| Weertman | 125 m | 456.7 km | 47737 |
| Tsai | 2 km | 437.9 km | 44996 |
| Tsai | 1 km | 440.0 km | 45238 |
| Tsai | 500 m | 442.9 km | 45700 |
| Tsai | 250 m | 444.1 km | 45899 |
| Tsai | 125 m | 444.1 km | 45889 |

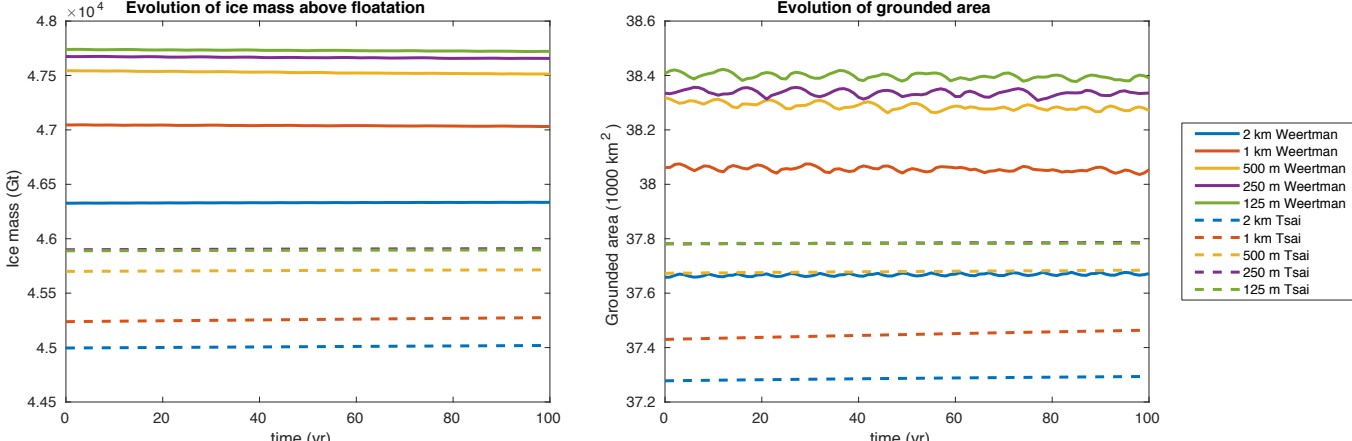

**Figure 4.** Evolution of ice volume above floatation (left) and grounded area (right) for Experiment 0 (steady-state case with no melt applied). Solid and dashed lines represent simulations with Weertman and Tsai friction respectively for resolutions of 2 km (blue), 1 km (red), 500 m (yellow), 250 m (purple), and 125 m (green). Results for 250 m and 125 m resolutions are superimposed for the Tsai friction.

Seroussi et al. (2014a)). The difference in modeled volume (see Table 1) between the 1 km and 500 m models is 1.02% and 1.05%, and the difference in grounded area is 0.61% and 0.62% respectively for the Weertman and Tsai friction laws. Differences between models at 500 m, 250 m, and 125 m resolution are all well below 1% (the curves for SS_Tsai_125m and SS_Tsai_250m are superimposed in Fig. 4). By comparison, the difference in volume above floatation and grounded area between the two friction laws at 125 m resolution is respectively of 3.9% and 1.6%.

**Table 2.** Change in volume above floatation ($\Delta$ VAF in Gt) in Experiment 1 for the Weerman (left) and Tsai (right) friction laws

| | Melt Parameterization (Weertman) | | | | | Melt Parameterization (Tsai) | | | |
|---|---|---|---|---|---|---|---|---|---|
| Resolution | NMP | FMP | SEM1 | SEM2 | Resolution | NMP | FMP | SEM1 | SEM2 |
| 2 km | -4137 | -5411 | -5210 | -5304 | 2 km | -5480 | -6692 | -6504 | -6576 |
| 1 km | -4272 | -4724 | -4637 | -4673 | 1 km | -6127 | -6454 | -6394 | -6417 |
| 500 m | -4246 | -4359 | -4331 | -4340 | 500 m | -6261 | -6333 | -6318 | -6324 |
| 250 m | -4225 | -4252 | -4244 | -4246 | 250 m | -6293 | -6315 | -6304 | -6305 |
| 125 m | -4196 | -4221 | -4213 | -4215 | 125 m | -6294 | -6307 | -6309 | -6311 |

Experiment 1 simulates the evolution of the glacier when ocean induced melt is applied under floating ice. The equation that governs the melt rate in this experiment provides limited melt close to the grounding line, as the water column thickness becomes smaller (see Eq. 3). Figure 5 shows the evolution of the ice volume above floatation for this experiment for the different sub-element melt parameterizations, mesh resolutions and the two friction laws. The volume above floatation lost (see also Table 2) varies between 4140 Gt and 6690 Gt for the EXP1_Weertman_2km_NMP and EXP1_Tsai_2km_FMP scenarios respectively. Experiments performed with the Tsai friction law show a larger mass loss (between 5480 and 6690 Gt over the 100-year period) than the ones performed with a Weertman friction law (between 4140 and 5410 Gt). The impact of the sub-element melt parameterization adopted, however, is more pronounced in the case of Weertman sliding law. The Tsai sliding law shows similar results for all sub-element parameterizations if the mesh resolution is 1 km and under, suggesting that any sub-element melt parameterization can be adopted in this case. Results performed at 2 km resolution all overestimate the mass loss, except when the NMP is adapted, which underestimate the mass loss. If the Weertman sliding law is applied, the results are strongly dependent on both the sub-element parameterization and the mesh resolution. SEM1, SEM2, and FMP behave very similarly, with mass loss being reduced as the resolution increases (from ~5400 Gt at 2 km resolution to ~4150 Gt at 250 m resolution). The difference between the runs becomes smaller as the mesh resolution increases, but the results are within 5% of the results obtained with a resolution of 125 m only for resolutions below 500 m. The NMP presents a completely different behavior, with results almost identical for all mesh resolutions for the Weertman sliding law (less than 150 Gt variation after 100 years). The runs relying on NMP underestimate the mass change for the Tsai friction law, with 650 Gt less mass loss for the EXP1_Tsai_2km_NMP compared to EXP1_Tsai_1km_NMP. During the experiment, the grounding line retreat in the centerline of the glacier varies between 40 and 55 km depending on the mesh resolution and the melt parameterization for the Weertman sliding law, and between 55 and 70 km for the Tsai sliding law, with larger retreats for the FMP, SMP1 and SMP2 at coarse resolution, and smaller retreats for FMP, SMP1 and SMP2 at fine resolution and NMP.

In Experiment 2, a large ice shelf melt rate of up to 30 m/yr is applied under the ice shelf, including close to the grounding line. Figure 6 and Table 3 show the results of this experiment for the different sub-element parameterizations, mesh resolutions, and the two sliding laws. The overall mass loss is similar to Experiment 1 and varies between 4110 Gt and 7590 Gt for

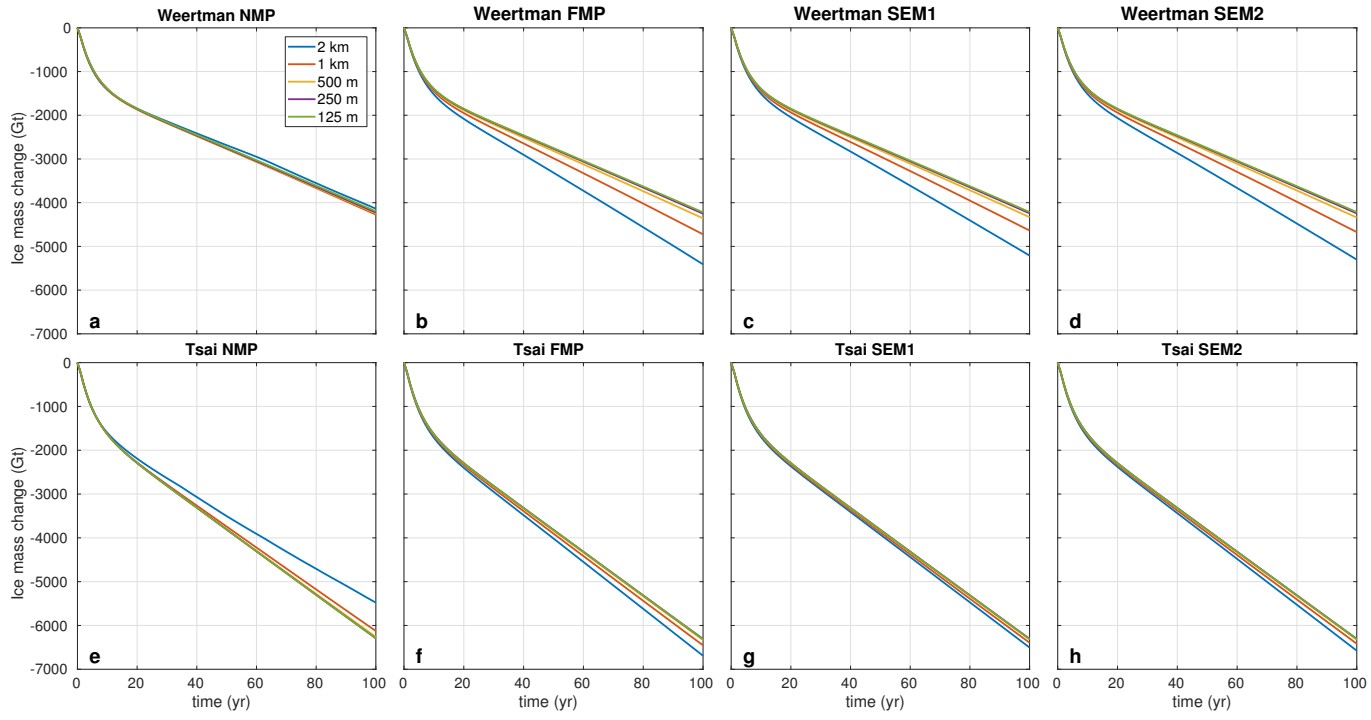

**Figure 5.** Evolution of ice volume above floatation in Experiment 1 for the NMP (a and e), FMP (b and f), SEM1 (c and g), and SEM2 (d and h), for the Weertman (a-d) and Tsai (f-h) friction laws. Each plot represents the evolution for the 5 mesh resolutions: 2 km (blue), 1 km (red), 500 m (yellow), 250 m (purple), and 125 m (green).

**Table 3.** Change in volume above floatation ($\Delta$ VAF in Gt) in Experiment 2 for the Weerman (left) and Tsai (right) friction laws

| Resolution | Melt Parameterization (Weertman) | | | | Resolution | Melt Parameterization (Tsai) | | | |
| --- | --- | --- | --- | --- | --- | --- | --- | --- | --- |
| | NMP | FMP | SEM1 | SEM2 | | NMP | FMP | SEM1 | SEM2 |
| 2 km | -4132 | -6536 | -5672 | -5644 | 2 km | -4943 | -7585 | -6614 | -6533 |
| 1 km | -4130 | -5895 | -5235 | -5188 | 1 km | -5150 | -7060 | -6365 | -6284 |
| 500 m | -4120 | -5289 | -4824 | -4775 | 500 m | -5374 | -6469 | -6034 | -5976 |
| 250 m | -4130 | -4890 | -4565 | -4523 | 250 m | -5474 | -6112 | -5846 | -5808 |
| 125 m | -4115 | -4748 | -4464 | -4428 | 125 m | -5510 | -6038 | -5812 | -5783 |

EXP1_Weertman_250m_NMP and EXP1_Tsai_2km_FMP scenarios respectively, with a larger ice loss for the Tsai friction law overall. The impact of mesh resolution and sub-element parameterization is more pronounced than in Experiment 1. At 2 km resolution, the difference in mass loss varies by 45% and 42% between NMP and FMP for both the Weertman and Tsai

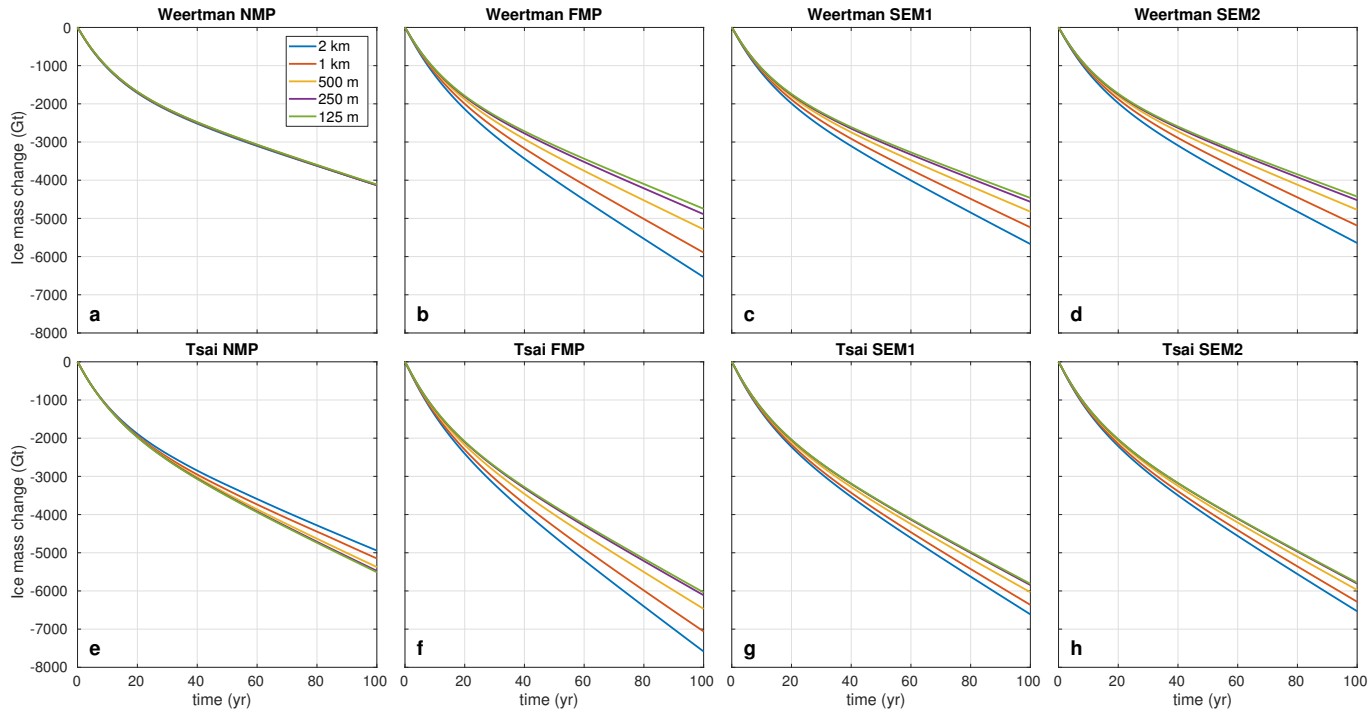

**Figure 6.** Evolution of ice volume above floatation in Experiment 2 for the NMP (a and e), FMP (b and f), SEM1 (c and g), and SEM2 (d and h), for the Weertman (a-d) and Tsai (f-h) friction laws. Each plot represents the evolution for the 5 mesh resolutions: 2 km (blue), 1 km (red), 500 m (yellow), 250 m (purple), and 125 m (green).

sliding laws, respectively. This spread is reduced as the mesh resolution increases, but a 125 m resolution is not sufficient to have similar results for NMP and FMP (14% and 9% difference between NMP and FMP at 125 m resolution for the Weertman and Tsai friction laws), suggesting that not all parameterizations have fully converged despite the level of mesh resolution. The SEM1 and SEM2 results are intermediate between FMP and NMP and behave similarly in all cases. Figure 6 also shows that

5    NMP is by far the least sensitive to mesh resolution for the Weertman sliding law, with, e.g., a mass change of only 20 Gt between EXP2_Weertman_2km_NMP and EXP2_Weertman_125m_NMP, whereas the difference reaches 1216 Gt between EXP2_Weertman_2km_SEM1 and EXP2_Weertman_125m_SEM1, and 1790 Gt between EXP2_Weertman_2km_FMP and EXP2_Weertman_125m_FMP. Results performed with the two sub-element melt parameterizations show a reduced dependence on mesh resolution. This improvement is not sufficient, however, to have accurate results with relatively coarse mesh

10   resolutions. The impact of mesh resolution and sub-element melt parameterization is more pronounced with the Weertman than the Tsai sliding friction law. Similarly to what was observed for Experiment 1, experiments performed with the Tsai friction law show less sensitivity to sub-element parameterization and mesh resolution than the Weertman friction law, except for NMP simulations that experience a mass loss reduced by 570 Gt over 100 years for the EXP2_Tsai_2km_NMP

compared to the EXP2_Tsai_125m_NMP. The difference in ice loss after 100 years between EXP2_Tsai_2km_FMP and the EXP2_Tsai_125m_FMP and between EXP2_Tsai_2km_SEM1 and EXP2_Tsai_125m_SEM1 is respectively reduced by 1000 and 800 Gt. During this experiment, the grounding line retreat in the centerline of the glacier varies between 33 and 63 km depending on the mesh resolution and the melt parameterization for the Weertman sliding law, and between 42 and 75 km for

the Tsai sliding law, with larger retreats for the FMP, SMP1 and SMP2 at coarse resolution, and smaller retreats for NMP and FMP, SMP1 and SMP2 at fine resolution.

## 5 Discussion

The results presented in this study show that the impact of sub-element melt parameterization and mesh resolution is different for the Weertman and Tsai friction laws. Models relying on Weertman sliding laws are more sensitive to the mesh resolution and

the type of sub-element melt parameterization than when a Tsai sliding law is employed. These conclusions are in agreement with the ones of Gladstone et al. (2017) on a flowline case. Figures 7 and 8 show the convergence of results with mesh resolution for the four sub-element mesh parameterizations. For the Weertman sliding law, the results vary by less than 2.0% for all the mesh resolutions regardless of the melt applied when NMP is used. Results using SEM1, SEM2, and FMP vary by at least one more order of magnitude, demonstrating that these parameterizations are more sensitive to mesh resolution than NMP in this

case. When a Tsai sliding law is used, the results vary depending on the amount of sub-ice shelf melt close to the grounding line. When melt rates converging towards zero close to the grounding line are applied, SEM1 and SEM2 converge slightly faster than FMP and NMP, and results within 5% of the 125 m resolution runs can be obtained for all sub-element parameterizations for mesh resolutions of 1 km or less. When large melt rates are applied close to the grounding line (Experiment 2), NMP converges the fastest but the behavior of SEM1 and SEM2 is close to NMP, with NMP underestimating the mass loss, while

SEM1 and SEM2 overestimate it. In all cases, SEM1 and SEM2 results are almost identical (similarly to what was observed for sub-element parameterization of basal friction, see Seroussi et al. (2014a)) and are intermediate between NMP and FMP. Differences between mass loss produced with NMP and FMP can be as large as about 50% for 2 km mesh resolution (see Fig. 6). This difference is reduced as the mesh resolution increases, but remains larger than 10% even at 125 m resolution (see Fig.6) for large melt rates. Using the FMP never produces the best convergence of results and overestimates the mass loss by a

factor of two in several cases, it should therefore be avoided. NMP shows the least dependence on mesh resolution, except for small melt rates close to the grounding line and a Tsai friction law (Figs. 7 and 8).

     To explain this behavior, one needs to look at the numerical implementation of the equations that are affected by melt. The ocean induced melt is only present as a right-hand side term in the mass transport equation:

$$\frac{\partial H}{\partial t} = -\nabla \cdot H\bar{\mathbf{v}} + \dot{a} - m_i \tag{5}$$

where $H$ is the ice thickness, $\bar{\mathbf{v}}$ is the depth averaged ice velocity, $\dot{a}$ is the surface mass balance. With the finite element method, $H$ is assumed to be a sum of nodal functions, and integrating basal melt, $m_i$, over partially floating elements will lead to a thinning at the grounded nodes of these elements that is inherent to the finite element method. In other words, applying melt

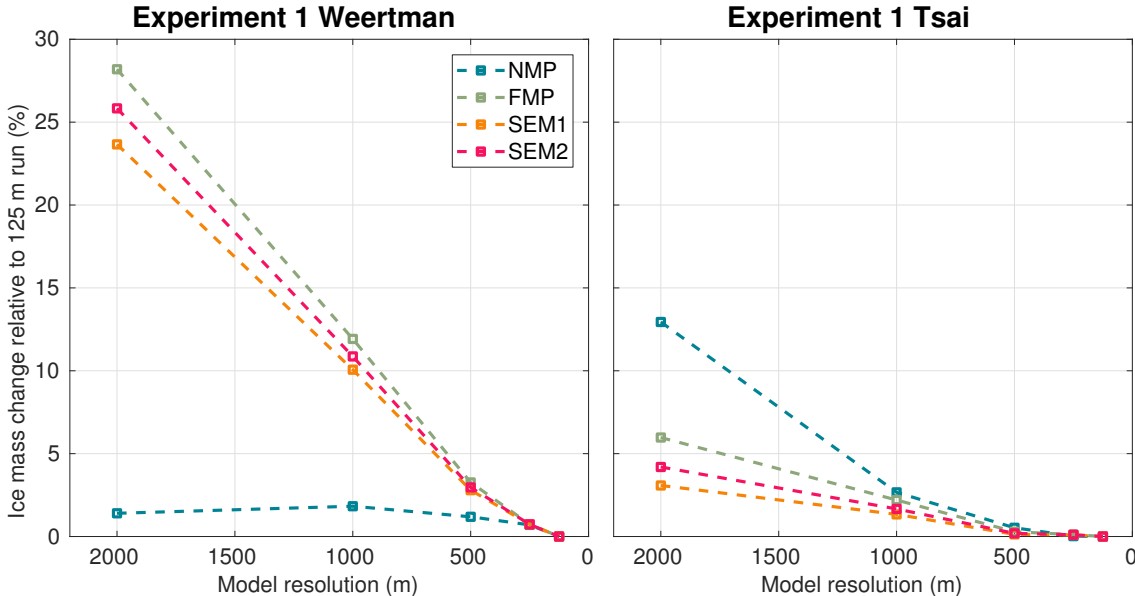

**Figure 7.** Convergence of ice volume above floatation at the end of Experiment 1 as a function of mesh resolution. Absolute error relative to the corresponding 125 mesh resolution results (same friction law and melt parameterization scheme) for the Weertman (a) and Tsai (b) friction laws for the NMP (blue), FMP (green), SEM1 (orange), and SEM2 (red) sub-element melt parameterizations.

in partially floating elements will induce a thinning upstream of the grounding line that is purely numerical, and the grounding line retreat will therefore be systematically overestimated. Using the no melt parameterization, no numerical thinning is applied to the grounded nodes of partially floating elements. Additional experiments, not shown here, confirm that even with a perfectly static marine ice sheet system (i.e., zero velocity at all time), the grounding line will artificially retreat, except for the NMP,

regarless of the numerical method adopted. This confirms that including some basal melting in partially floating elements or cells will overestimate grounding line retreat or may lead to grounding line retreats in cases where the grounding line should theoretically not retreat. This happens despite the fact that the basal melt rate applied through SEM2 is exact (i.e., basal melting is applied only under the floating part of the domain) and independent of mesh resolution, while NMP and FMP overestimate and underestimate the total amount of basal melting, respectively.

Unlike what has been recommended for sub-element parameterizations of basal friction at the grounding line (e.g., Pattyn et al., 2006; Vieli and Payne, 2005; Feldmann et al., 2014; Seroussi et al., 2014a), using a sub-element *melt* parameterization does therefore not guarantee an improvement compared to simulations that do not include such implementations, and does not necessarily relax the requirements of mesh resolutions. This is especially true when large melt rates are applied in the vicinity of the grounding line and for the Weertman sliding law. Many simulations in the Amundsen Sea Sector of West Antarctica

(e.g. Favier et al., 2014; Joughin et al., 2014; Seroussi et al., 2014b) applied large melt rates in this region, consistently with observations (Dutrieux et al., 2013). A previous model study performed with NMP and SEM1 on this region showed extreme

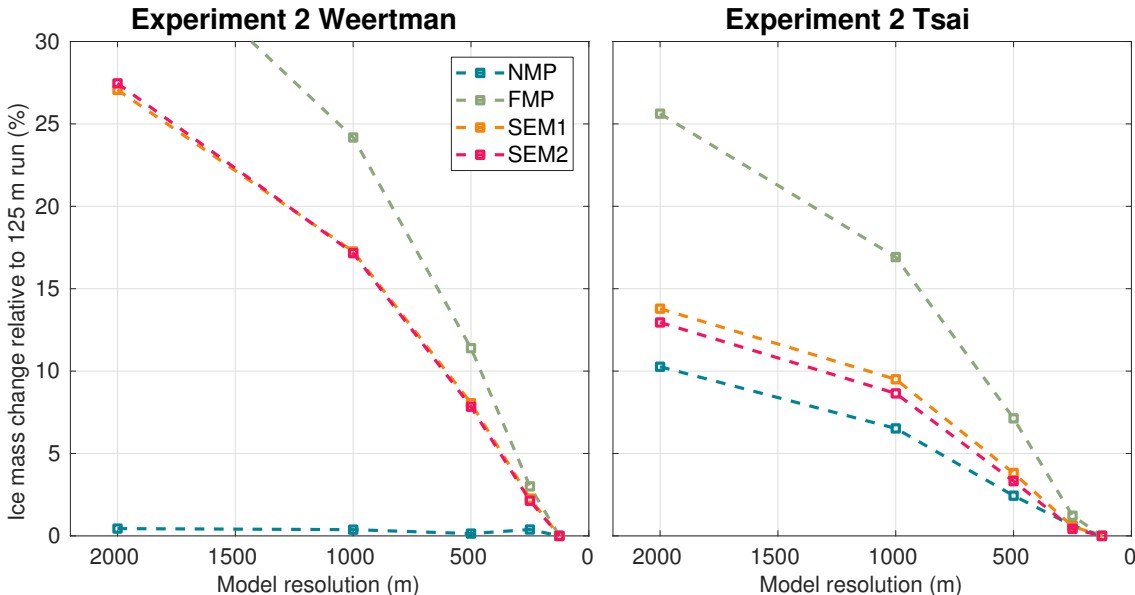

**Figure 8.** Convergence of ice volume above floatation at the end of Experiment 2 as a function of mesh resolution. Absolute error relative to the corresponding 125 mesh resolution results (same friction law and melt parameterization scheme) for the Weertman (a) and Tsai (b) friction laws for the NMP (blue), FMP (green), SEM1 (orange), and SEM2 (red) sub-element melt parameterizations.

differences even over 100 years, and a potential collapse of Thwaites Glacier in less than 100 years for large melt rate scenarios (Arthern and Williams, 2017). Our study sheds light on this problem, as the SEM1 was probably under-resolved, leading to an overestimation of grounding line retreat.

In this study, we only considered mesh resolutions that are 2 km or less. However, large scale simulations of the Antarctic
ice sheet typically rely on significantly coarser resolutions (e.g., Golledge et al., 2015; DeConto and Pollard, 2016; Pollard et al., 2015), especially when performing long term simulations. In this case, using the FMP, SEM1, and SEM2 will always lead to large overestimates in the amount of mass loss and even collapse of entire regions if large melt rates are applied close to the grounding line or if experiment scenarios include large melt rates in these regions, for both Weertman and Tsai sliding laws. As mentioned in previous studies (e.g., Cornford et al., 2016; Gladstone et al., 2017), quantifying the impact of mesh
resolution on model results is therefore extremely important in this case in order to provide reliable estimates of uncertainties in ice sheet mass loss over the coming decades and centuries. This is especially important when simulating the collapse of marine terminating glaciers resting on retrograde bed slope that are sensitive to the Marine Ice Sheet Instability (MISI, Weertman (1974)), as such an instability would be potentially simulated several centuries too early if ice shelf melt rates are applied on partially floating elements (Arthern and Williams, 2017; Golledge et al., 2015).
The results presented here were all performed on simulations that experience grounding line retreat and no grounding line advance. As most glaciers around the world are experiencing sustained retreat in response to climate change, cases of

grounding line advance are less common. The numerical scheme or resolution needed to correctly reproduce grounding line advance are however different than those needed to accurately capture grounding line retreat: Gladstone et al. (2017) showed that convergence was even worse in the case of grounding line advance. Convergence tests are even more critical to perform in such a case.

Grounding lines are constantly migrating, not only on long time scales due to changes in oceanic or atmospheric conditions, but also over short time scales with tides (e.g. Gudmundsson, 2007; Le Meur et al., 2014; Padman et al., 2018). Observations show that melting in the grounding zones is complex and tidal motion probably involves complex melt rate patterns changing on tidal time scales as grounding line advances and retreats, and tidal flexure pumps ocean water in the grounding zone (Walker et al., 2013). This process could lead to more complicated patterns than the ones used in this study, assuming that the ice shelf is in hydrostatic equilibrium. However, such processes remain poorly understood, additional studies are required to better evaluate them, and should not be used as a justification for numerical model inaccuracy.

All the simulations performed in this study are based on the two-dimensional SSA. We expect, however, the results to be qualitatively similar for other stress balance approximations that determine the grounding line position based on the hydrostatic equilibrium, as melt rates in partially floating elements are treated in a similar way regardless of the stress balance approximation. Using a Stokes flow line model, Gladstone et al. (2017) demonstrate a similar greater dependence of model results when large melt rates are applied close to the grounding line and the need for stricter resolution requirements. Simulations performed with three dimensional higher-order (Pattyn, 2003) or L1L2 (Hindmarsh, 2004) models should however generally experience lower changes in these cases, as previous studies showed that SSA models tend to respond more quickly than models including vertical shear (Pattyn et al., 2013; Pattyn and Durand, 2013).

# 6 Conclusions

In this study we investigate the impact of the numerical implementation of ice shelf melt rates immediately downstream of the grounding line. We compare several sub-element parameterizations that (1) do not apply any melt over partially floating elements, (2) apply basal melt over the entire partially floating elements, or (3) apply some melt over partially floating elements. Simulations are performed with different mesh resolutions for two experiments with small and large melt rates close to the grounding line, and for a Weertman and a Tsai sliding laws. Our results demonstrate that, for limited melt rates in the order of 1 m/yr close to the grounding line, all sub-element melt parameterizations behave similarly for resolutions lower than 1 km and 500 m respectively for the Tsai and Weertman friction laws. For large melt rates in the order of 30 m/yr just downstream of the grounding line, however, models based on varying resolutions and sub-element melt rates behave differently. Both (2) and (3) overestimate the mass loss and resolutions well below 500 m are needed, while (1) shows a behavior that is less dependent on the mesh resolution. These results were performed using the finite element method, but can be extrapolated to other numerical methods, such as the finite element and finite volume methods. As continental scale simulations of Antarctica typically use resolutions of several kilometers in the grounding line region, we therefore recommend models not to apply ice shelf melt rates in over the entire partially floating elements and to carefully assess the impact of mesh resolution and sub-element melt parameterization on all simulation results.

*Acknowledgements.* The research was carried out at the Jet Propulsion Laboratory, California Institute of Technology, under a contract with the National Aeronautics and Space Administration. Funding was provided by grants from the NASA Cryospheric Science and Jet Propulsion Laboratory Research Technology and Development Programs. We thank S. Cornford and the two reviewers R. Gladstone and D. Martin for their constructive comments that improved the clarity of the paper.

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
