# Peer review of "Representation of basal melting at the grounding line in ice flow models"

_The Cryosphere, 2018_

## Referee Comment (RC1) · R. Gladstone (Referee) · 12 Jul 2018

The paper addresses the impact of schemes for handling sub-shelf melting in elements containing a section of grounding line in ice sheet models. The main points are that fairly strict resolution requirements might need to be imposed in order to provide a converged result in the presence of high melt rates near the grounding line, and that applying melt over partially grounded elements when resolution is not sufficiently fine is likely to give an overestimate of retreat rates and mass loss for a retreating grounding line. The paper is clearly written, the experiments simple and to the point, and the figures show the scientific content very clearly. This is a useful and timely contribution to the ice sheet modelling community. Anyone carrying out marine ice sheet modelling needs to be aware of the main points made by this paper.

[Figure]

No advance experiments were carried out. Of course one cannot conduct advance experiments by starting from a steady state without melting and then imposing melting, but starting from a steady state with high melting and then reducing the melt rate is perfectly viable. If we consider the grounding line convergence issue due to the basal resistance change across the grounding line, some models/parameterisations give better convergence in advance and some in retreat. There may be a similar issue with the melt problem. If you look at the bottom left panel of Fig 5 of Gladstone et al 2017 (also TC) you can see that we observed worse convergence in advance than in retreat in the presence of basal melting with no parameterisation (albeit with a different sliding relation to the ones used here). Of course in the current climate we're more interested in retreat than advance but temporary advance could occur as part of a larger retreat pattern (see also Jong et al TCD 2017 (now accepted for TC), and Torsten Albrecht's work on overshoot (work from last year, not sure if it is published yet, but you can contact him if you want to know more)). The addition of advance experiments would enhance this paper, but then again the paper is worth publishing as is and needs to be brought to the attention of other modellers. So I do not have a strong preference whether advance experiments should be added at his point. But at least add some brief discussion of the implications. Why does Weertman sliding with no melt parameterisation show such wonderful convergence? Perhaps it would be terrible in advance?

Why should experiments with the Tsai sliding law show less sensitivity to melt parameterisations than experiments with the Weertman sliding law? Is it something to do with the different geometries – Weertman having thicker ice and steeper slopes close to the grounding line?

Specific comments

Line 4. add -> adds

Line 19. Yields -> leads?

Line 16. "impact to" -> "impact on"

Line 21. "except if specified" -> "except where specified"

Line 32. Can you state what temperature this corresponds to?

Line 10. The connectivity is between the subglacial hydrologic system and the ocean. Just saying "bed" allows the possibility of a dry bed, which cannot support sliding.

Lines 8-11. This means the initial steady state is always approached through advance. This is a good design for retreat perturbations (I also have a paper coming out in TCD in the next few days that discusses multiple steady states and design of grounding line experiments). But you have not said how steady state is defined. You state that all spin up simulations are at least 50ka, which is good and should suffice for a robust starting point, but can you also add a statement about steady state, e.g. "dVAF/dt is less than xx km^3/a in all cases" or similar? [edit: I see you discuss Exp0 further down to look closer at steady state. So ignore my comment about quantifying steady state here, but add a line like "Achievement of steady-state is analysed through Experiment 0 below"]

Lines 12-17. Note that "-Xm below sea level" contains a double negative (because the minus sign and the word "below" imply direction) and technically would mean Xm above sea level. You should say "Xm below sea level" or "-Xm relative to sea level (where X is positive in the upward direction)".

Lines 12-17. Please add the equation for this melt parameterisation, since you've shown one for the other parameterisation.

Interactive
comment

Line 2. "resolution" -> "resolutions"

Line 2. Those numbers don't look right to me, looking at the plot. Did you get the Tsai and Weertman laws mixed up in either the text of the plot?

Line 5. "small oscillations with minimal amplitude" -> "small oscillations"

Lines 2-4. This is quite similar to our TC paper (Gladstone et al 2017): the first melt param used here (your exp 1) is similar to the water column scaling we used. Scaling the melt to zero as the GL is approached reduces the resolution dependency.

Line 6. "the type sub-element" -> "the sub-element" or "the type of sub-element"

Line 7. "a mass loss" -> "mass loss"

Line 5. "why" -> "whereas"

Lines 6-7. Experiment names are repeated when they are clearly supposed to be different experiments.

line 11. I think the point here is not that the melt rates are small generally, but that they approach zero as the grounding line is approached (due to the water column scaling).

Page 10 line 13 to page 11 line 5. Do you think this problem is specific to the finite element method? Steph Cornford essentially predicted these results based on theoretical reasoning a couple of years ago (this was in a conversation, don't think he published anything like this). He said he would expect any parameterisation that allows melt on the last grounded grid point to overestimate retreat and to give worse convergence than a scheme that only applied melt to the first floating grid point. He mainly uses

finite difference or finite volume methods.

Line 13. "even other" -> "even over"

Lines 11-17. Well, yes, but such processes could well mean that the melt parameterisations are actually closer to reality than NMP, though of course parameterising a tidal grounding zone should also not be resolution dependent.

Line 17. Isn't there a PISM paper that does exactly that – using the grounding zone concept as a justification for inaccuracy. . . not sure if it is constructive to point the finger by citing it though. . .

Lines 18-23. One of the main points is similar to Gladstone et al 2017 (also TC), which used a Stokes flow model: that the convergence is worse, and resolution requirements are stricter, for the case of high melt close to the grounding line. The importance of vertical shear probably depends on choice of sliding law – vertical shear should have a larger impact when using Weertman than with one of the sliding relations featuring a grounded transition zone.

Line 28. "large amount of" -> "large"

line 29. "for a Weertman and a Tsai sliding laws"

lines 29 and 31. Please indicate roughly what "large" and "small" mean here, for the benefit of people who just look at the pictures and read the conclusion!

Figures.

Fig 4. Right panel y-axis label. Minor formatting issue. Large gap in km^2.

Fig 4. I presume the Tsai purple line is hidden behind the green one? i.e. perfect convergence at 250m? You should state this in the caption or readers might think the

purple line is missing. I find it confusing switching between Figures 5 and 7 because the colours have completely different meaning. Could you manage different line types in Fig 7 instead of different colours? Or if you want to stick with colours, can you make the colours different from those in Fig 5? I instinctively see the blue line in Fig 7 and think "ah that's the 2km resolution"...

What I am missing from all Figures is a way to compare the converged result across different melt parameterisations. Of course the converged result should be the same across all melt parameterisations for a given sliding law. But this is hard to compare in Fig 5 because they all have separate sub plots, and Fig 7 shows relative differences. I'm not sure the best way to show this, perhaps a new figure or just a table... and of course it may be complicated by the oscillations in the Weertman case. I don't view this as essential, just "would be nice".

I found myself wanting to look at Fig 7 while looking at Fig 5. Maybe you should swap around Figs 6 and 7 and refer to the convergence plot a bit earlier in the text? Just a thought – I am not insisting on this.

---

## Referee Comment (RC2) · D. Martin (Referee) · 24 Jul 2018

**Review of "Representation of basal melting at the grounding line in ice flow models"**

Dan Martin

July 24, 2018

This work explores the convergence and accuracy characteristics of a set of choices in representing subshelf melting near marine ice sheet grounding lines. Since grounding lines often fall in the interior of computational grid cells, modelers are presented with a decision on how to represent subshelf melting in partially-grounded cells. One can either restrict melt in the model to cells which are entirely floating, include full representations of subshelf melting in all partially- and fully-grounded cells, or use some sort of scheme which reduces the model melt in partially-grounded cells to account for the fact that such cells are only partially floating. Existing model results in the literature employ the full range of these approaches, with unknown impacts on the model projections.

In this work, the authors employ four schemes to represent melt near grounding lines: (1) a scheme in which melt forcing is only applied to fully-floating cells (their "NMP"), (2) a scheme in which melt is applied fully to all cells which are even partially floating ("FMP"), and (3) two schemes which attempt to represent a subgrid-scale distribution of the melt forcing (in which forcing will tend to zero as the floating area in the cell tends to zero) ("SMP"). They apply these choices to an idealized marine ice stream problem (MISMIP+) with two melt parameterizations designed to test two different regimes. The experiments are carried out over a range of model resolutions, designed to examine the convergence behavior of each scheme. They find that schemes which apply melting to partially-grounded cells (both the "FMP" and "SMP") tend to over-represent ice sheet response, particularly at coarse resolutions, while the no-melt ("NMP") scheme tends to under-predict ice sheet response, while also displaying better accuracy and convergence properties. Therefore, their conclusion is that one should use schemes which don't apply melt to partially-grounded cells, particularly at the coarse resolutions typically used for full-ice-sheet-scale models.

Given the importance of subshelf-melt forcing to marine ice sheet dynamics and its relevance to projections of ice sheet contributions to sea level rise, and the fact that many studies predict large melt rates near the grounding lines, this work is a very important step toward understanding how to incorporate subshelf melt into modeling efforts in an accurate way. The paper itself is well-constructed, clearly-written, and was a pleasure to read. The authors present

a convincing explanation of their results, and their conclusion is well-supported by their results. I fully support publication, after a few minor fixes.

**Specific notes:**

1. (p1, line 5): "which ultimately add..." – add → adds

2. (p2, line 20): It would be nice if you would also specify boundary conditions here to give a better feel for the problem without having to look up the citation for those unfamiliar with the Asay-Davis paper.

3. Figure 3 (and accompanying text): It would be helpful to see a convergence plot like figures 7&8 for the steady-state initial condition (or the results of Experiment 0) in order to see how the model itself is converging independent of the melt behavior (to better place the melt convergence results into context).

4. Experiments 1 and 2: How far does the GL (centerline) retreat in these experiments? It's useful to have some context relative to the coarse mesh spacing. (i.e. if it's only retreating $O(10)$ 2km cells, that could be relevant, particularly in terms of how smooth the NMP and FMP parameterizations would appear to the ice sheet (since they're discontinuous in time, while the SMP ones are continuous as the GL retreats).

5. (Figure 4): It's apparent that the 2km results aren't even in the asymptotic convergence regime. (just an observation, which would be clearer if there was a figure like I suggest in (3) above)

6. (p7, line 6): "type sub-element" → "type of sub-element"

7. (p7, line 7): "with a mass loss" → "with mass loss"

8. (p7, line 10): Are the different experiments all converging to the same solution? I think they are, but you never actually say that, and it's hard to tell exactly from the figures given their size)

9. (p8, line 5): "why the difference" → "while the difference"

10. (Figures 5 and 6): I find the mesh-independence of the Weertman NMP case surprising, particularly for experiment 2, since you're potentially omitting a fair bit of melt near the GL. Any idea why this case looks that way?

11. (p.10, line 3): "overestime" → "overestimate"

12. (p.10, line 8): "mass by a factor" → "mass loss by a factor"

13. (p.10, line 9): "the grounding" → "the grounding line"

14. (Figures 7 and 8): I *think* you're referencing each scheme to its own 125m result, which is problematic given that you observe that not all of them are fully resolved at 125m for experiment 2 (this will tend to underestimate the real error being made here for those cases, since figures 7-8 imply that all of the models have error which goes to 0 at 125 m.). I'd suggest instead referencing them all to the same baseline result. Since the 125m NMP run appears to be converged, I'd suggest using that result as the baseline value to compare all of the other results to. Or, you could run a single 62.5m NMP run (which should be very close to the 125 NMP) and use *that* as the baseline solution. Regardless, you should clarify what the reference choice is.

15. (Figures 7 and 8): It might help to mention that you're plotting the abs(error) in these plots, which winds up being the negative of the actual difference for the NMP case. (I spent some time trying to figure out why the NMP line in the Experiment 1 Tsai figure was above the FMP line, until I remembered the sign difference)

16. (p. 11, line 5): This conclusion likely holds broadly for any model which applies melt forcing over the entire cell (including finite-difference and finite-volume approaches), not just simply the ISSM FEM model. For example, I'd expect the finite-volume BISICLES to behave the same way.

17. (p. 11, line 8): guaranty → guarantee

18. (p. 11, line 13): "even other" → "even over"

19. (p. 12, line 6): This is an important point which can't be repeated enough. You could cite Cornford et al (2016) here, which also makes the point about the necessity to quantify or clarify the effects of mesh resolution; both this work and that one provide a template for how to go about doing it (mesh convergence study).

**References:**

20. (p. 14, line 11). My name is spelled incorrectly

21. (p. 14, line 29). "West Antarctica"

22. (p. 15, line 3): "andvvan" → "and V van"

---

## Author Comment (AC1) · 9 Aug 2018

Dear Pr. Gagliardini,

We are grateful to the two reviewers, who have provided helpful and insightful comments to our manuscript. All of their remarks have been taken into account. Please find attached our responses to the reviewers and a description of how we changed the manuscript. The new version of the manuscript now includes several new tables with simulations results in order to more easily compare the different runs. The new manuscript is included in the supplement along with our responses, and differences with the previous version are highlighted.

Best regards,

[Figure]

Helene Seroussi

Please also note the supplement to this comment:
https://www.the-cryosphere-discuss.net/tc-2018-117/tc-2018-117-AC1-
supplement.pdf

---

## Author Response (AR1)

**1 Reviewer #1 (R. Gladstone)**

The paper addresses the impact of schemes for handling sub-shelf melting in elements containing a section of grounding line in ice sheet models. The main points are that fairly strict resolution requirements might need to be imposed in order to provide a converged result in the presence of high melt rates near the grounding line, and that applying melt over partially grounded elements when resolution is not sufficiently fine is likely to give an overestimate of retreat rates and mass loss for a retreating grounding line. The paper is clearly written, the experiments simple and to the point, and the figures show the scientific content very clearly. This is a useful and timely contribution to the ice sheet modelling community. Anyone carrying out marine ice sheet modelling needs to be aware of the main points made by this paper.

No advance experiments were carried out. Of course one cannot conduct advance experiments by starting from a steady state without melting and then imposing melting, but starting from a steady state with high melting and then reducing the melt rate is perfectly viable. If we consider the grounding line convergence issue due to the basal resistance change across the grounding line, some models/parameterisations give better convergence in advance and some in retreat. There may be a similar issue with the melt problem. If you look at the bottom left panel of Fig 5 of Gladstone et al 2017 (also TC) you can see that we observed worse convergence in advance than in retreat in the presence of basal melting with no parameterisation (albeit with a different sliding relation to the ones used here). Of course in the current climate we're more interested in retreat than advance but temporary advance could occur as part of a larger retreat pattern (see also Jong et al TCD 2017 (now accepted for TC), and Torsten Albrecht's work on overshoot (work from last year, not sure if it is published yet, but you can contact him if you want to know more)). The addition of advance experiments would enhance this paper, but then again the paper is worth publishing as is and needs to be brought to the attention of other modellers. So I do not have a strong preference whether advance experiments should be added at his point. But at least add some brief discussion of the implications. Why does Weertman sliding with no melt parameterisation show such wonderful convergence? Perhaps it would be terrible in advance?

We thank Rupert Gladstone for his detailed review and constructive comments. The question of numerical treatment needed to accurately represent grounding line advance is indeed an important one that deserves being investigated as well. One complication is that as melt needs to be applied to grow the glacier during the initial steady-state in this case, the initial state is going to be impacted by the choice of melt parameterization. For this reason, we decided to add a paragraph in the discussion to discuss this question and refer to previous studies that are mentioned here. We discuss the convergence of Weertman NMP further down.

Why should experiments with the Tsai sliding law show less sensitivity to melt

parameterisations than experiments with the Weertman sliding law? Is it something to do with the different geometries? Weertman having thicker ice and steeper slopes close to the grounding line?

We think that when a Weertman sliding law is used, as the slopes are steeper close to the grounding line, the thinning is more localized to the area just upstream of the ice shelf, while Tsai friction favors rapid spread of thinning further inland of the glacier.

Specific comments Page 1 Line 4. add  $\rightarrow$  adds Done

Line 19. Yields  $\rightarrow$  leads?

Done

Page 2 Line 16. "impact to"  $\rightarrow$  "impact on"

Done

Line 21. "except if specified"  $\rightarrow$  "except where specified"

Done

Line 32. Can you state what temperature this corresponds to?

This corresponds to a temperature of about -9°C. Added

Page 3 Line 10. The connectivity is between the subglacial hydrologic system and the ocean. Just saying "bed" allows the possibility of a dry bed, which cannot support sliding.

Clarified

Page 4 Lines 8-11. This means the initial steady state is always approached through advance. This is a good design for retreat perturbations (I also have a paper coming out in TCD in the next few days that discusses multiple steady states and design of grounding line experiments). But you have not said how steady state is defined. You state that all spin up simulations are at least 50ka, which is good and should suffice for a robust starting point, but can you also add a statement about steady state, e.g. "dVAF/dt is less than xx km3/a in all cases" or similar? [edit: I see you discuss Exp0 further down to look closer at steady state. So ignore my comment about quantifying steady state here, but

add a line like "Achievement of steady-state is analysed through Experiment 0 below"]

Done

Page 5 Lines 12-17. Note that "-Xm below sea level" contains a double negative (because the minus sign and the word "below" imply direction) and technically would mean Xm above sea level. You should say "Xm below sea level" or "-Xm relative to sea level (where X is positive in the upward direction)".

The double negative was removed.

Lines 12-17. Please add the equation for this melt parameterisation, since you've shown one for the other parameterisation.

We added the melt equation for Experiment 2.

Page 6 Line 2. "resolution"  $\rightarrow$  "resolutions"

Done

Line 2. Those numbers don't look right to me, looking at the plot. Did you get the Tsai and Weertman laws mixed up in either the text of the plot?

The numbers are indeed the right numbers (the confusion probably comes from the most advanced position being on the right panel). We added a table detailing the initial conditions (grounding line position and volume above floatation) to clarify that.

Line 5. "small oscillations with minimal amplitude"  $\rightarrow$  "small oscillations"

Done

Page 7 Lines 2-4. This is quite similar to our TC paper (Gladstone et al 2017): the first melt param used here (your exp 1) is similar to the water column scaling we used. Scaling the melt to zero as the GL is approached reduces the resolution dependency.

We added a sentence referencing this study in the discussion.

Line 6. "the type sub-element"  $\rightarrow$  "the sub-element" or "the type of sub-element"

Done

Line 7. "a mass loss"  $\rightarrow$  "mass loss"

Done

Page 8 Line 5. "why"  $\rightarrow$  "whereas"

Done

Lines 6-7. Experiment names are repeated when they are clearly supposed to be different experiments.

We remplaced the names with the right experiment names.

Page 9 line 11. I think the point here is not that the melt rates are small generally, but that they approach zero as the grounding line is approached (due to the water column scaling).

This is correct (and what is stated in the previous sentence), we clarified this sentence.

Page 10 line 13 to page 11 line 5. Do you think this problem is specific to the finite element method? Steph Cornford essentially predicted these results based on theoretical reasoning a couple of years ago (this was in a conversation, don't think he published anything like this). He said he would expect any parameterisation that allows melt on the last grounded grid point to overestimate retreat and to give worse convergence than a scheme that only applied melt to the first floating grid point. He mainly uses finite difference or finite volume methods.

This is indeed not specific to the finite element method, and we should expect similar results with other numerical methods. We added a sentence to generalize the results to other methods in the discussion.

Page 11 Line 13. "even other"  $\rightarrow$  "even over"

Done

Page 12 Lines 11-17. Well, yes, but such processes could well mean that the melt parameterisations are actually closer to reality than NMP, though of course parameterising a tidal grounding zone should also not be resolution dependent.

We think this highlights that we need to better understand what happens close to the grounding line and in very shallow water columns, with the complexity of adding tides. However, guessing what would happen in the presence of tides is well beyond the scope of this manuscript.

Line 17. Isn't there a PISM paper that does exactly that ? using the grounding zone concept as a justification for inaccuracy... not sure if it is constructive to point the finger by citing it though...

We think parameterization of melt "at the grounding line" is an important point to study to raise awareness in the community and avoid redoing the same mistakes, not to emphasize what was not done wrong in the past.

Lines 18-23. One of the main points is similar to Gladstone et al 2017 (also TC), which used a Stokes flow model: that the convergence is worse, and resolution requirements are stricter, for the case of high melt close to the grounding line. The importance of vertical shear probably depends on choice of sliding law vertical shear should have a larger impact when using Weertman than with one of the sliding relations featuring a grounded transition zone.

We added a reference to the similar conclusion in Gladstone et al. (2017).

Line 28. "large amount of"  $\rightarrow$  "large"

Done

line 29. "for a Weertman and a Tsai sliding laws"

Done

lines 29 and 31. Please indicate roughly what "large" and "small" mean here, for the benefit of people who just look at the pictures and read the conclusion!

Done

Figures. Fig 4. Right panel y-axis label. Minor formatting issue. Large gap in  $\rm km^2.$

Done

Fig 4. I presume the Tsai purple line is hidden behind the green one? i.e. perfect convergence at 250m? You should state this in the caption or readers might think the purple line is missing. I find it confusing switching between Figures 5 and 7 because the colours have completely different meaning. Could you manage different line types in Fig 7 instead of different colours? Or if you want to stick with colours, can you make the colours different from those in Fig 5? I instinctively see the blue line in Fig 7 and think "ah that's the 2km resolution"...

Done: added comment in the caption to say that purple and green line are superimposed. Also changed the colors for figures 7 and 8.

What I am missing from all Figures is a way to compare the converged result across different melt parameterisations. Of course the converged result should be the same across all melt parameterisations for a given sliding law. But this is hard to compare in Fig 5 because they all have separate sub plots, and Fig 7 shows relative differences. I'm not sure the best way to show this, perhaps a new figure or just a table... and of course it may be complicated by the oscillations in the Weertman case. I don't view this as essential, just "would be nice".

We added two tables detailing the ice loss for experiments 1 and 2 to provide an easy comparison. We show ice mass loss values (similar to figures 5 and 6) and not total ice as the oscillations in the initial steady-state and different initial values between the resolutions could be confusing.

I found myself wanting to look at Fig 7 while looking at Fig 5. Maybe you should swap around Figs 6 and 7 and refer to the convergence plot a bit earlier in the text? Just a thought I am not insisting on this.

We decided to keep the order of the figures, as it was consistent with the reasoning in our results and discussion.

**2 Reviewer #2 (D. Martin)**

This work explores the convergence and accuracy characteristics of a set of choices in representing subshelf melting near marine ice sheet grounding lines. Since grounding lines often fall in the interior of computational grid cells, modelers are presented with a decision on how to represent subshelf melting in partially-grounded cells. One can either restrict melt in the model to cells which are entirely floating, include full representations of subshelf melting in all partially- and fully-grounded cells, or use some sort of scheme which reduces the model melt in partially-grounded cells to account for the fact that such cells are only partially floating. Existing model results in the literature employ the full range of these approaches, with unknown impacts on the model projections.

In this work, the authors employ four schemes to represent melt near grounding lines: (1) a scheme in which melt forcing is only applied to fully-floating cells (their "NMP"), (2) a scheme in which melt is applied fully to all cells which are even partially floating (FMP), and (3) two schemes which attempt to represent a subgrid-scale distribution of the melt forcing (in which forcing will tend to zero as the floating area in the cell tends to zero) ("SMP"). They apply these choices to an idealized marine ice stream problem (MISMIP+) with two melt parameterizations designed to test two different regimes. The experiments are carried out over a range of model resolutions, designed to examine the convergence behavior of each scheme. They find that schemes which apply melting to partially-grounded cells (both the "FMP" and "SMP") tend to over-represent ice sheet response, particularly at coarse resolutions, while the no-melt ("NMP") scheme tends to under-predict ice sheet response, while also displaying better accuracy and convergence properties. Therefore, their conclusion is that one should use schemes which don't apply melt to partially-grounded cells, particularly at the coarse resolutions typically used for full-ice-sheet-scale models.

Given the importance of subshelf-melt forcing to marine ice sheet dynamics and its relevance to projections of ice sheet contributions to sea level rise, and the fact that many studies predict large melt rates near the grounding lines, this work is a very important step toward understanding how to incorporate subshelf melt into modeling efforts in an accurate way. The paper itself is wellconstructed, clearly-written, and was a pleasure to read. The authors present a convincing explanation of their results, and their conclusion is well-supported by their results. I fully support publication, after a few minor fixes.

We thank Dan Martin for his detailed review and insightful comments.

Specific notes: 1. (p1, line 5): "which ultimately add..."  $add \rightarrow adds$

Done

2. (p2, line 20): It would be nice if you would also specify boundary conditions here to give a better feel for the problem without having to look up the citation for those unfamiliar with the Asay-Davis paper.

Done

3. Figure 3 (and accompanying text): It would be helpful to see a convergence plot like figures 7 and 8 for the steady-state initial condition (or the results of Experiment 0) in order to see how the model itself is converging independent of the melt behavior (to better place the melt convergence results into context).

Figure 7 and 8 show the convergence of the change in volume above floatation, and there is no change in volume above floatation in Experiment 0 (as it is just to test the steady-state). To be able to compare the initial steady-states, we added a table (Table 1) with the initial volumes above floatation and grounding line positions, as well as two tables showing the change in volume above floatation for experiments 1 and 2 so that the numbers can be easily compared.

4. Experiments 1 and 2: How far does the GL (centerline) retreat in these experiments? It's useful to have some context relative to the coarse mesh spacing. (i.e. if it's only retreating O(10) 2km cells, that could be relevant, particularly in terms of how smooth the NMP and FMP parameterizations would appear to the ice sheet (since they're discontinuous in time, while the SMP ones are continuous as the GL retreats)).

The grounding line retreat varies between 33 and 75 km, depending on the sliding law, melt experiment and melt parameterization. We added a couple of sentences in the results to discuss this retreat.

5. (Figure 4): It's apparent that the 2km results aren't even in the asymptotic convergence regime. (just an observation, which would be clearer if there was a figure like I suggest in (3) above)

Thanks for pointing this out.

6. (p7, line 6): "type sub-element"  $\rightarrow$  "type of sub-element"

Done

7. (p7, line 7): "with a mass loss"  $\rightarrow$  "with mass loss"

Done

8. (p7, line 10): Are the different experiments all converging to the same solution? I think they are, but you never actually say that, and it's hard to tell exactly from the figures given their size)

We added two tables as suggested by reviewer 1 so that it's easier to see if the solutions converge towards the same state.

9. (p8, line 5): "why the difference"  $\rightarrow$  "while the difference"

Done

10. (Figures 5 and 6): I find the mesh-independence of the Weertman NMP case surprising, particularly for experiment 2, since you're potentially omitting a fair bit of melt near the GL. Any idea why this case looks that way?

We were also surprised by this result, so we ran the simulations several times and the results are robust. We don't have a detailed explanation, but we think this is caused by the Weertman sliding law not being dependent on the effective pressure, so the thinning is limited to the ice shelf and does not propagate far inland. As the friction remains unchanged in all cases (except where the grounding line retreats), the changes in driving stress between the different resolutions are similar enough to cause similar changes in the simulations.

11. (p.10, line 3): "overestime"  $\rightarrow$  "overestimate"

Done

- 12. (p.10, line 8): "mass by a factor"  $\rightarrow$  "mass loss by a factor" Done
- 13. (p.10, line 9): "the grounding"  $\rightarrow$  "the grounding line"

**Done**

14. (Figures 7 and 8): I \*think\* you're referencing each scheme to its own 125 m result, which is problematic given that you observe that not all of them are fully resolved at 125 m for experiment 2 (this will tend to underestimate the real error being made here for those cases, since figures 7-8 imply that all of the models have error which goes to 0 at 125 m.). I'd suggest instead referencing them all to the same baseline result. Since the 125 m NMP run appears to be converged, I'd suggest using that result as the baseline value to compare all of the other results to. Or, you could run a single 62.5m NMP run (which should be very close to the 125 NMP) and use \*that\* as the baseline solution. Regardless, you should clarify what the reference choice is.

We do reference each scheme to its 125 m resolution result. We think this has a limited impact, as the results between the different are not very different at this resolution for experiment 1 (see table 2). For experiment 2 (see table 3), the results between NMP and the other schemes are large enough that it is not entirely clear if we have reached a perfect convergence at 125 m. We added some clarification in the figure captions.

15. (Figures 7 and 8): It might help to mention that you're plotting the abs(error) in these plots, which winds up being the negative of the actual difference for the NMP case. (I spent some time trying to figure out why the NMP line in the Experiment 1 Tsai figure was above the FMP line, until I remembered the sign difference)

Agreed, we clarified the caption.

16. (p. 11, line 5): This conclusion likely holds broadly for any model which applies melt forcing over the entire cell (including finite-difference and finite-volume approaches), not just simply the ISSM FEM model. For example, I'd expect the finite-volume BISICLES to behave the same way.

We generalized the conclusion.

17. (p. 11, line 8): guaranty  $\rightarrow$  guarantee

Done

18. (p. 11, line 13): "even other"  $\rightarrow$  "even over"

Done

19. (p. 12, line 6): This is an important point which can't be repeated enough. You could cite Cornford et al (2016) here, which also makes the point about the necessity to quantify or clarify the effects of mesh resolution; both this work and that one provide a template for how to go about doing it (mesh convergence study).

Good point, we added references to previous studies who mention the importance of quantifying mesh resolution.

References: 20. (p. 14, line 11). My name is spelled incorrectly

Sorry we mispelled your name. Fixed

- 21. (p. 14, line 29). "West Antarctica" Done
- 22. (p. 15, line 3): "and vvan"  $\rightarrow$  "and V van" Done

[revised manuscript text omitted]

---

## Editor Decision (ED1)

**Representation of basal melting at the grounding line in ice flow models**

Hélène Seroussi[1] and Mathieu Morlighem[2]

[1]Jet Propulsion Laboratory - California Institute of Technology, Pasadena, CA 91109, USA
[2]Department of Earth System Science, University of California Irvine, Irvine, CA 92697, USA

*Correspondence to:* Helene Seroussi (Helene.Seroussi@jpl.nasa.gov)

**Abstract.** While a lot of attention has been given to the numerical implementation of grounding lines and basal friction in the grounding zone, little has been done about the impact of the numerical treatment of ocean-induced basal melting in this region. Several strategies are currently being employed in the ice sheet modeling community, and the resulting grounding line dynamics may differ strongly, which ultimately adds significant uncertainty to the projected contribution of marine ice sheets to sea level rise. We investigate here several implementations of basal melt parameterization on partially floating elements in a finite element framework, based on the Marine Ice Sheet-Ocean Model Intercomparison Project (MISOMIP) setup: (1) melt applied only to entirely floating elements, (2) melt applied over the entire elements that are crossed by the grounding line, and (3) melt integrated partially over the floating portion of a finite element using two different sub-element integration methods. All methods converge towards the same state when the mesh resolution is fine enough. However, (2) and (3) will systematically overestimate the rate of grounding line retreat in coarser resolutions, while (1) converges faster to the solution in most cases. The differences between sub-element parameterizations are exacerbated for experiments with large melting rates in the vicinity of the grounding line and for a Weertman sliding law. As most real-world simulations use horizontal mesh resolutions of several hundreds of meters at best, and large melt rates are generally present close to the grounding lines, we recommend using (1) to avoid overestimating the rate of grounding line retreat.

**1 Introduction**

Basal melt under floating ice tongues is important as it is one of the main factors driving the current increase in ice discharge in West Antarctica (e.g. Pritchard et al., 2012). Changes in basal melt impact ice shelf thickness, and thinning leads to a reduction of ice shelf buttressing, thereby leading to an acceleration of the ice streams feeding it. This acceleration is responsible for the dynamic thinning of the ice upstream of the grounding line, eventually leading to grounding line retreat, which causes to a further increase in ice speed, and therefore ice discharge. Accurate representation of ice shelf ocean-induced melt in ice flow models is therefore critical. This remains an active field of research as observations of basal melt remain scarce, and new parameterizations are starting to emerge (Lazeroms et al., 2018; Reese et al., 2017).

Over the past decade, the ice sheet modeling community has made tremendous progress in terms of representation of grounding line dynamics in ice sheet models. Model intercomparisons have shown that lateral stress and high mesh resolution (below

**Résumé des commentaires sur tc-2018-117-manuscript-version5.pdf**

**Page : 1**

Nombre : 1      Auteur : ogagliardini  Sujet : Texte surligné      Date : 26/08/2018 13:01:32

this is based on the convergence results. Physically, (3) sounds better founded?

[revised manuscript text omitted]

Nombre : 1        Auteur : ogagliardini  Sujet : Texte surligné        Date : 26/08/2018 13:13:18

Figure

**3 Experiments**

We first run every configuration to a steady-state ice stream without any melt. The initial ice thickness is equal to 1 m and the ice stream grows over several tens of thousands of years (at least 50,000 years) in response to surface mass balance accumulation, while no basal melting is applied under floating ice. This steady-state is therefore independent of the sub-element basal melt parameterization applied. Convergence of the solution to the steady-state is discussed in the analysis of Experiment 0 in Section 4.

Starting from this steady-state, three transient experiments with varying ice shelf basal melting conditions are performed for a period of 100 years. In Experiment 0, no basal melting is applied under floating ice, similar to the steady-state initialization of the model. Experiment 0 is therefore mainly designed to check the initial steady-state. Basal melting is applied under floating ice in Experiment 1 and Experiment 2, and we assess the impact of the melt parameterization, model resolution and sliding laws on the glacier evolution. Experiment 1 is similar to the MISMIP+ *Ice1r* experiment in Asay-Davis et al. (2016): basal melting varies spatially and represents a balance between the latent heat of melting and a parameterized ocean turbulent heat flux:

$$
m_i = \Omega \tanh \left( \frac{H_c}{H_{c0}} \right) \max \left( z_0 - z_d, 0 \right) \tag{3}
$$

with $\Omega$ a coefficient equal to 0.2 $\text{yr}^{-1}$, $H_c$ the water column thickness, $z_d$ the ice shelf basal elevation, $z_0$ the depth above which the melt rate is equal to zero (100 m), and $H_{c0}$ a constant equal to 75 m (see also equations (12)-(17) in Asay-Davis et al. (2016) for the derivation of this parameterization).

Experiment 2 is based on a basal melt under floating ice that varies linearly with depth, with a maximum melt magnitude of 30 m/yr in the deepest part, where the ice base is at or below 500 m below sea level, and that linearly decreases to 0 m/yr melt for ice base equal to 50 m below sea level. There is therefore no melt when the ice base is above 50 m below sea level:

$$
m_i = \begin{cases} 0 \text{ m/yr}, & \text{if } z_d > -50 \text{ m} \\ -1/15 \left( z_d + 50 \right) \text{ m/yr}, & \text{if } -500 < z_d < -50 \text{ m} \\ 30 \text{ m/yr}, & \text{if } z_d < -500 \text{ m} \end{cases} \tag{4}
$$

with $z_d$ the ice shelf basal elevation. This experiment simulates ice shelves resting in warm waters, similarly to what has been observed in the Amundsen or Bellingshausen sea areas (e.g. Dutrieux et al., 2013; Rignot et al., 2013) and used in previous modeling experiments (e.g. Favier et al., 2014; Joughin et al., 2014; Seroussi et al., 2014b, 2017).

Experiments 0, 1 and 2 are all run for 100 years. We use the following convention to refer to the different experiments. For the steady-state (SS) and Experiment 0, names are as follows: EXP_sliding_resolution, where EXP is the number of the experiment (SS or EXP0), the sliding refers to the sliding law (Weertman or Tsai), and 'resolution' is the mesh resolution (2 km, 1 km, 500 m, 250 m, or 125 m), e.g., EXP0_Weertman_250m. For Experiment 1 and Experiment 2, the names are similar: EXP_sliding_resolution_SEM, except that we add SEM, the sub-element melt parameterization at the grounding line

**Page : 5**

Nombre : 1          Auteur : ogagliardini   Sujet : Texte surligné          Date : 26/08/2018 13:12:56
Section

Nombre : 1          Auteur : ogagliardini   Sujet : Texte surligné          Date : 26/08/2018 13:12:56
Section

[Figure]

**Figure 3.** Steady-state grounding line positions for the Weertman [1] (left) and Tsai (right) friction law for the 2 km (blue), 1 km (red), 500 m (yellow), 250 m (purple) and 125 m (green) mesh resolutions

(NMP, FMP, SEM1 or SEM2), e.g., EXP1_Weertman_250m_SEM1, as the results of these simulations now depend on the sub-element melt parameterization adopted.

**4 Results**

Figure 1 shows the initial steady-state configuration for SS_Weertman_125m. Its geometry is shown in Fig. 1a, and the velocity

5 and grounding line in Fig. 1b. The grounding line position varies between 458 km in the centerline of the glacier and 528 km on its sides; the ice velocity is maximum at the ice front, reaching 1012 m/yr. This configuration is comparable to previous results based on the same geometry (Gudmundsson et al., 2012; Gudmundsson, 2013; Asay-Davis et al., 2016). The mesh resolution and the type of basal sliding law both impact the grounding line position as shown in Fig. 3. The grounding line position on the glacier centerline varies between 438 km for SS_Tsai_2km and 458 km for SS_Weertman_125m, with a larger spread between

10 the different resolutions for the Tsai friction law (9.6 km) than for the Weertman friction law (6.2 km) (Fig. 3 and [2]Tab. 1).

Experiment 0 is mostly designed to ensure that the model has reached a steady-state, as no melt is applied, similar to the initial steady-state. The ice mass above floatation (Fig. 4a) remains constant over the 100-year simulation for the 10 configurations, while the grounded ice area (Fig. 4b) experiences small oscillations with small oscillations, especially for the Weertman sliding law. Such oscillations, that average to zero change in the grounded area over time, have been noted by Asay-

15 Davis et al. (2016) and are orders of magnitude smaller than the changes simulated in Experiment 1 and Experiment 2. Figure 4 confirms that sub-kilometer resolution is needed to accurately capture the grounding line positions, similarly to what has been suggested by previous studies (e.g., Vieli and Payne (2005); Gladstone et al. (2010); Pattyn et al. (2012, 2013); Feldmann et al. (2014); Seroussi et al. (2014a)). The difference in modeled volume (see [3]Table 1) between the 1 km and 500 m models

**Page : 6**

| | | | |
|---|---|---|---|
| **T** Nombre : 1 | Auteur : ogagliardini | Sujet : Texte surligné | Date : 26/08/2018 13:19:56 |

mention that the 250m and 125m solution for Tsai are superimposed.

| | | | |
|---|---|---|---|
| **T** Nombre : 2 | Auteur : ogagliardini | Sujet : Texte surligné | Date : 26/08/2018 13:21:14 |

Table

| | | | |
|---|---|---|---|
| **T** Nombre : 3 | Auteur : ogagliardini | Sujet : Texte surligné | Date : 26/08/2018 13:23:28 |

Table

**Table 1.** Steady-state grounding line position in the glacier centerline and volume above floatation (VAF)

| Friction law | Resolution | GL($y = 40$ km) | VAF (Gt) |
| --- | --- | --- | --- |
| Weertman | 2 km | 448.0 km | 46327 |
| Weertman | 1 km | 452.8 km | 47044 |
| Weertman | 500 m | 456.3 km | 47540 |
| Weertman | 250 m | 456.6 km | 47674 |
| Weertman | 125 m | 456.7 km | 47737 |
| Tsai | 2 km | 437.9 km | 44996 |
| Tsai | 1 km | 440.0 km | 45238 |
| Tsai | 500 m | 442.9 km | 45700 |
| Tsai | 250 m | 444.1 km | 45899 |
| Tsai | 125 m | 444.1 km | 45889 |

[Figure]

**Figure 4.** Evolution of ice volume above floatation (left) and grounded area (right) for Experiment 0 (steady-state case with no melt applied). Solid and dashed lines represent simulations with Weertman and Tsai friction respectively for resolutions of 2 km (blue), 1 km (red), 500 m (yellow), 250 m (purple), and 125 m (green). Results for 250 m and 125 m resolutions are superimposed for the Tsai friction.

is 1.02% and 1.05%, and the difference in grounded area is 0.61% and 0.62% respectively for the Weertman and Tsai friction laws. Differences between models at 500 m, 250 m, and 125 m resolution are all well below 1% (the curves for SS_Tsai_125m and SS_Tsai_250m are superimposed in Fig. 4). By comparison, the difference in volume above floatation and grounded area between the two friction laws at 125 m resolution is respectively of 3.9% and 1.6%.

**Page : 7**

 Nombre : 1         Auteur : ogagliardini  Sujet : Texte surligné         Date : 26/08/2018 13:32:26
the legend could appear only in the left panel as it is the same for both panels.

 Nombre : 2         Auteur : ogagliardini  Sujet : Texte surligné         Date : 26/08/2018 13:24:19
in

**Table 2.** Change in volume above floatation ($\Delta$ VAF in Gt) in Experiment 1 for the Weerman (left) and Tsai (right) friction laws

| Resolution | Melt Parameterization (Weertman) | | | | Resolution | Melt Parameterization (Tsai) | | | |
|---|---|---|---|---|---|---|---|---|---|
| | NMP | FMP | SEM1 | SEM2 | | NMP | FMP | SEM1 | SEM2 |
| 2 km | -4137 | -5411 | -5210 | -5304 | 2 km | -5480 | -6692 | -6504 | -6576 |
| 1 km | -4272 | -4724 | -4637 | -4673 | 1 km | -6127 | -6454 | -6394 | -6417 |
| 500 m | -4246 | -4359 | -4331 | -4340 | 500 m | -6261 | -6333 | -6318 | -6324 |
| 250 m | -4225 | -4252 | -4244 | -4246 | 250 m | -6293 | -6315 | -6304 | -6305 |
| 125 m | -4196 | -4221 | -4213 | -4215 | 125 m | -6294 | -6307 | -6309 | -6311 |

Experiment 1 simulates the evolution of the glacier when ocean induced melt is applied under floating ice. The equation that governs the melt rate in this experiment provides limited melt close to the grounding line, as the water column thickness becomes smaller (see Eq. 3). Figure 5 shows the evolution of the ice volume above floatation for this experiment for the different sub-element melt parameterizations, mesh resolutions and the two friction laws. The volume above floatation lost (see also Table 2) varies between 4140 Gt and 6690 Gt for the EXP1_Weertman_2km_NMP and EXP1_Tsai_2km_FMP scenarios respectively. Experiments performed with the Tsai friction law show a larger mass loss (between 5480 and 6690 Gt over the 100-year period) than the ones performed with a Weertman friction law (between 4140 and 5410 Gt). The impact of the sub-element melt parameterization adopted, however, is more pronounced in the case of Weertman sliding law. The Tsai sliding law shows similar results for all sub-element parameterizations if the mesh resolution is 1 km and under, suggesting that any sub-element melt parameterization can be adopted in this case. Results performed at 2 km resolution all overestimate the mass loss, except when the NMP is adapted, which underestimate the mass loss. If the Weertman sliding law is applied, the results are strongly dependent on both the sub-element parameterization and the mesh resolution. SEM1, SEM2, and FMP behave very similarly, with mass loss being reduced as the resolution increases (from $\sim$5400 Gt at 2 km resolution to $\sim$4150 Gt at 250 m resolution). The difference between the runs becomes smaller as the mesh resolution increases, but the results are within 5% of the results obtained with a resolution of 125 m only for resolutions below 500 m. The NMP presents a completely different behavior, with results almost identical for all mesh resolutions for the Weertman sliding law (less than 150 Gt variation after 100 years). The runs relying on NMP underestimate the mass change for the Tsai friction law, with 650 Gt less mass loss for the EXP1_Tsai_2km_NMP compared to EXP1_Tsai_1km_NMP. During the experiment, the grounding line retreat in the centerline of the glacier varies between 40 and 55 km depending on the mesh resolution and the melt parameterization for the Weertman sliding law, and between 55 and 70 km for the Tsai sliding law, with larger retreats for the FMP, SMP1 and SMP2 at coarse resolution, and smaller retreats for FMP, SMP1 and SMP2 at fine resolution and NMP.

In Experiment 2, a large ice shelf melt rate of up to 30 m/yr is applied under the ice shelf, including close to the grounding line. Figure 6 and Table 3 show the results of this experiment for the different sub-element parameterizations, mesh resolutions, and the two sliding laws. The overall mass loss is similar to Experiment 1 and varies between 4110 Gt and 7590 Gt for

**Page : 8**

Nombre : 1      Auteur : ogagliardini   Sujet : Texte surligné      Date : 26/08/2018 13:26:41
Table

Nombre : 2      Auteur : ogagliardini   Sujet : Texte surligné      Date : 26/08/2018 13:35:50
Table

[revised manuscript text omitted]

**T** Nombre : 1      Auteur : ogagliardini   Sujet : Texte surligné      Date : 26/08/2018 13:47:00

this part should be discussed a bit more, as again, the SEM2 implementation seems the more physically founded one? With the NMP, total melt is underestimated and the total melt is dependant on the mesh resolution, which should not be the case for the SEM2 parametrization?

**T** Nombre : 2      Auteur : ogagliardini   Sujet : Texte surligné      Date : 26/08/2018 13:44:17

what is v here? The depth averaged velocity?

**T** Nombre : 3      Auteur : ogagliardini   Sujet : Texte surligné      Date : 26/08/2018 13:49:43

I don't fully agree with your conclusion especially if you only look at Tsai results. And as many works recommend not to use anymore a Weertman friction law in the vicinity of GL, I think you should moderate this conclusion.

that convergence was even worse in the case of grounding line advance. Convergence tests are even more critical to perform in such a case.

Grounding lines are constantly migrating, not only on long time scales due to changes in oceanic or atmospheric conditions, but also over short time scales with tides (e.g. Gudmundsson, 2007; Le Meur et al., 2014; Padman et al., 2018). Observations show that melting in the grounding zones is complex and tidal motion probably involves complex melt rate patterns changing on tidal time scales as grounding line advances and retreats, and tidal flexure pumps ocean water in the grounding zone (Walker et al., 2013). This process could lead to more complicated patterns than the ones used in this study, assuming that the ice shelf is in hydrostatic equilibrium. However, such processes remain poorly understood, additional studies are required to better evaluate them, and should not be used as a justification for numerical model inaccuracy.

All the simulations performed in this study are based on the two-dimensional SSA. We expect, however, the results to be qualitatively similar for other stress balance approximations that determine the grounding line position based on the hydrostatic equilibrium, as melt rates in partially floating elements are treated in a similar way regardless of the stress balance approximation. Using a Stokes flow line model, Gladstone et al. (2017) demonstrate a similar greater dependence of model results when large melt rates are applied close to the grounding line and the need for stricter resolution requirements. Simulations performed with three dimensional higher-order (Pattyn, 2003) or L1L2 (Hindmarsh, 2004) models should however generally experience lower changes in these cases, as previous studies showed that SSA models tend to respond more quickly than models including vertical shear (Pattyn et al., 2013; Pattyn and Durand, 2013).

**6 Conclusions**

In this study we investigate the impact of the numerical implementation of ice shelf melt rates immediately downstream of the grounding line. We compare several sub-element parameterizations that (1) do not apply any melt over partially floating elements, (2) apply basal melt over the entire partially floating elements, or (3) apply some melt over partially floating elements. Simulations are performed with different mesh resolutions for two experiments with small and large melt rates close to the grounding line, and for a Weertman and a Tsai sliding laws. Our results demonstrate that, for limited melt rates in the order of 1 m/yr close to the grounding line, all sub-element melt parameterizations behave similarly for resolutions lower than 1 km and 500 m respectively for the Tsai and Weertman friction laws. For large melt rates in the order of 30 m/yr just downstream of the grounding line, however, models based on varying resolutions and sub-element melt rates behave differently. Both (2) and (3) overestimate the mass loss and resolutions well below 500 m are needed, while (1) shows a behavior that is less dependent on the mesh resolution. These results were performed using the finite element method, but can be extrapolated to other numerical methods, such as the finite element and finite volume methods. As continental scale simulations of Antarctica typically use resolutions of several kilometers in the grounding line region, [1] we therefore recommend models not to apply ice shelf melt rates in partially floating elements and [2] to carefully assess the impact of mesh resolution on their simulation results.

*Acknowledgements.* The research was carried out at the Jet Propulsion Laboratory, California Institute of Technology, under a contract with the National Aeronautics and Space Administration. Funding was provided by grants from the NASA Cryospheric Science and Jet

**Page : 14**

 Nombre : 1     Auteur : ogagliardini   Sujet : Texte surligné     Date : 26/08/2018 14:27:35

to be reconsidered or at least better discussed in the manuscript

 Nombre : 2     Auteur : ogagliardini   Sujet : Texte surligné     Date : 26/08/2018 14:27:07

this is a good point

Propulsion Laboratory Research Technology and Development Programs. We thank S. Cornford, I. Gladstone and D. Martin for their constructive comments that improved the clarity of the paper.

Nombre : 1          Auteur : ogagliardini   Sujet : Texte surligné          Date : 26/08/2018 14:28:01
should ne mentioned that both were reviewers.

---

## Author Response (AR2)

**1 Editor comments**

Dear Hélène and Mathieu,

Thanks for this new version of your paper and your careful reply to both reviewers. Both of them have positively evaluated the work presented in our paper and made pertinent remarks to improve the manuscript. I think that your paper can be accepted for publication in The Cryosphere, but after a careful reading I still have few remarks and found some typos that should be corrected for the final version.

My main remarks is on the strong conclusion of the paper that recommand to use the NMP parametrization. I think this should be discussed a bit further, based on the following points: - Among the 4 parametrizations, the most physical one is the SEM2: it is the one that reproduce "exactly" the melt function you want to apply everywhere on the domain. None of the other do that (you might think adding a figure showing how the melt evolve in the vicinity of the GL for Exps 1 and 2 and the 4 melt parametrizations, and compare it to the mathematical function of melt). - Only SEM2 and SEM1 are mass conservative for the total melt (total melt is independent of the mesh discretization). The NMP will give less total melt as mesh size increase, and it is the reverse for FMP. In other words, the forcing (or total melt below the ice shelf) is dependant on the mesh size for NMP and FMP, not for SEM1 and SEM2. - The conclusion that NMP works better than the 3 others is obvious for the Weertman friction, but this is less clear when looking at Tsai results (see below). Isn't it a bit hazardous to build the conclusion on the friction law that has been shown not to be adapted for the vicinity of the GL? At least, this should be discussed. - For Tsai friction law and Exp 1, I would recommand to use SEM1 or SEM2, not NMP, if only looking at the mesh dependency. For Exp 2 and Tsai, the dependency to mesh is very comparable for NMP, SEM1 and SEM2, but the ice mass change are much larger for SEM1 and SEM2 than NMP (and SEM1 and SEM2 gives similar results). Which mass changes are the most correct? Based on the previous remark, I will be more confident in the results given by SEM2 (confirmed by those obtained with SEM1) than those of NMP.

In short, I think you can draw different conclusions from your work and you should be very careful in bringing the whole community to use an inappropriate melt parametrization for bad reasons.

In the supplementary document you will find an annotated manuscript with some typos that you should correct in the revised version.

Best Regards, Olivier

We thank the editor for its carefull reading of our manuscript. We modified the abstract, discussion and conclusions to nuanced our conclusions on the sub-element parameterization that should be adopted, so that they better reflect the results found in this study.

p.1: this is based on the convergence results. Physically, (3) sounds better founded?

(3) is indeed what sounds more physical at first as the correct amount of melt is applied. However, the numerical treatment of partially floating elements lead to dynamical changes that can overestimate grounding line retreats. We nuanced this statement as results can be different for the low melt case with the Tsai friction law.

**p.2: should not be here but beow just before presenting the BC at the bed.**

Done

**p.3: Figure**

Done

**p.3: Section**

Done

**p.6: mention that the 250m and 125m solution for Tsai are superimposed.**

Done

**p.6: Table**

Done

**p.7: the legend could appear only in the left panel as it is the same for both panels.**

Done

**p.7: in**

Done

**p.8: Table**

Done

**p.11: Is not satisfying the good wording?**

We rephrased this sentence.

**p.11: Figs.**

Done

p.11: averaged?

Done

p.12: this part should be discussed a bit more, as again, the SEM2 implementation seems the more physically founded one? With the NMP, total melt is underestimated and the total melt is dependant on the mesh resolution, which should not be the case for the SEM2 parametrization?

We added this point to the discussion.

p.12: what is v here? The depth averaged velocity?

Added the definition of v.

p.12: I don't fully agree with your conclusion especially if you only look at Tsai results. And as many works recommend not to use anymore a Weertman friction law in the vicinity of GL, I think you should moderate this conclusion.

We kept this sentence as we think that "using a sub-element melt parameterization does therefore not guarantee an improvement compared to simulations that do not include such implementations" is a nuanced statement designed to encourage modelers to look critically at their results instead of applying the same melt parameterization scheme for all situations (which as seen in this study does not exist) without analysing the impact on simulation results and chosing an appropriate numerical scheme.

p.14: to be reconsidered or at least better discussed in the manuscript

We nuanced this conclusion.

p.14: this is a good point

Thanks.

p.15: should ne mentioned that both were reviewers.

Done

[revised manuscript text omitted]